# Gut Microbiome Modulation and Health Benefits of a Novel Fucoidan Extract from *Saccharina latissima*: A Double-Blind, Placebo-Controlled Trial

**DOI:** 10.3390/microorganisms13071545

**Published:** 2025-06-30

**Authors:** Gissel Garcia, Josanne Soto, Carmen Valenzuela, Mirka Bernal, Jesús Barreto, María de la C. Luzardo, Raminta Kazlauskaite, Neil Waslidge, Charles Bavington, Raúl de Jesús Cano

**Affiliations:** 1Pathology Department, Hermanos Ameijeiras Clinical and Surgical Hospital, Calle San Lázaro No 701, Esq. a Belascoaín, Centro Habana, La Habana 10400, Cuba; gisselgarcia2805@gmail.com; 2Clinical Laboratory Department, Clinical Hospital “Hermanos Ameijeiras”, Calle San Lázaro No 701, Esq. a Belascoaín, Centro Habana, La Habana 10400, Cuba; josanne.soto@infomed.sld.cu (J.S.); mirkab@infomed.sld.cu (M.B.); 3Institute of Cybernetics, Mathematics and Physics Havana University, La Habana 10400, Cuba; carmenvalenzuelasilva@gmail.com; 47 Nutrition Department, Clinical Hospital “Hermanos Ameijeiras”, Calle San Lázaro No 701, Esq. a Belascoaín, Centro Habana, La Habana 10400, Cuba; barreto.penie@gmail.com; 5Protein Studies Center, Biology Faculty, Havana University, Calle 25 Entre Calle J e I No 455 Plaza, La Habana 10400, Cuba; mcluzardo28@gmail.com; 6Oceanium, Ltd., Malin House, Suite10 The European Marine Science Park, Dunbeg, Oban PA37 1SZ, UK; raklauskaite@oceanium.co.uk (R.K.); nwaslidge@oceanium.co.uk (N.W.); charlie@oceanium.co.uk (C.B.); 7Biological Sciences Department, California Polytechnic State University, San Luis Obispo, CA 93407, USA; 8Academia de Ciencias de Cuba, La Habana 12400, Cuba

**Keywords:** *Saccharina latissima*, fucoidan, clinical trial, microbiome, SCFA, intestinal transit

## Abstract

This randomized, double-blind, placebo-controlled, three-arm clinical trial evaluated the effects of a proprietary bioactive fucoidan-rich extract derived from *Saccharina latissima* (SLE-F) on gut microbial composition and function in healthy adults. The objective of the study was to assess the potential of SLE-F to beneficially modulate the gut microbiome, with this paper specifically reporting on microbial diversity, taxonomic shifts, and functional pathway outcomes. Ninety-one participants received either a low dose (125 mg), high dose (500 mg), or placebo twice daily for ninety days. The primary endpoint was the microbiome composition assessed via 16S rRNA sequencing (V3–V4 region), with secondary outcomes including surveys, adverse event monitoring, and clinical evaluations. High-dose supplementation resulted in dose-dependent improvements in the microbial diversity; increased abundance of beneficial taxa, including *Bifidobacterium*, *Faecalibacterium*, and *Lachnospiraceae*; and reductions in inflammation-associated taxa, such as *Enterobacteriaceae* and *Pseudomonadota*. A functional pathway analysis showed enhancement in short-chain fatty acid biosynthesis and carbohydrate metabolism. The low-dose group showed modest benefits, primarily increasing *Bifidobacterium*, with limited functional changes. In vitro colonic simulations further demonstrated a dose-dependent increase in short-chain fatty acids and postbiotic metabolite production following SLE-F exposure. SLE-F was well tolerated, with only mild, nonspecific adverse events reported. These findings support the potential of SLE-F as a safe and effective microbiome-modulating agent, warranting further study of the long-term use and synergy with dietary interventions.

## 1. Introduction

The gut microbiome, comprising a highly dynamic consortium of trillions of microorganisms, exerts a fundamental influence on human physiology by modulating metabolic pathways, immune system function, and cardiovascular regulation [1,2]. A diverse and balanced microbiota contributes to gut barrier integrity, inflammation regulation, and improved metabolic outcomes, while dysbiosis is associated with various conditions, including inflammatory bowel disease (IBD), metabolic disorders, and cardiovascular diseases [3,4,5]. Emerging evidence underscores the importance of microbial metabolites—such as short-chain fatty acids (SCFAs)—in shaping host physiology, influencing systemic inflammation, and regulating cardiometabolic risk factors [6,7,8].

Dietary interventions that leverage prebiotic substrates to selectively enhance beneficial microbial activity offer a promising strategy for improving human health. Ocean Actives H+ (SLE-F), a proprietary bioactive extract of the brown seaweed *Saccharina latissima*, is rich in mixed fucoidan species, proteins, and polyphenols. Fucoidans, a family of sulfated polysaccharides derived primarily from brown seaweed, have been increasingly recognized for their prebiotic properties, selectively promoting the growth of beneficial gut bacteria and supporting metabolic and immune functions [9,10,11,12]. Recent studies have demonstrated that fucoidan’s bioactivity is closely linked to its molecular characteristics, including sulfation degree, molecular weight, and monosaccharide composition, which affect microbial fermentability and downstream host responses [13,14].

SLE-F has demonstrated the ability to stimulate microbial fermentation and SCFA production, positioning it as a potential therapeutic agent for gut health. In a human clinical trial evaluating fucoidan as an adjunct during *Helicobacter pylori* eradication therapy, fucoidan supplementation helped preserve gut microbial diversity and SCFA-producing taxa, such as *Bifidobacterium* and *Faecalibacterium*, and was associated with improved gastrointestinal tolerance [11].

Preclinical studies utilizing ProDigest’s (Gent, Belgium) colonic simulation system and short-term Caco-2/THP1 leaky gut model provided mechanistic insights into SLE-F’s impact on the gut microbiota composition, metabolic activity, and gut barrier function. While fermentation of SLE-F did not directly enhance the gut barrier integrity under inflammatory conditions, it selectively enriched beneficial bacterial taxa, including *Butyricicoccus* and *Lachnospiraceae*, and exhibited a favorable metabolomic signature associated with cardiometabolic health. Complementary murine models of diet-induced obesity and colitis have shown that fucoidan can enrich SCFA-producing taxa, such as *Akkermansia*, while attenuating the gut permeability and systemic inflammation, supporting its relevance for inflammatory and metabolic disorders [15,16].

These findings provided the foundation for a three-arm, double-blind, placebo-controlled clinical trial aimed at evaluating the impact of SLE-F on gut microbial composition over a three-month intervention period. The study included two treatment groups—LD (125 mg/capsule, twice daily) and HD (500 mg/capsule, twice daily)—alongside a placebo (microcrystalline cellulose 125 mg/capsule).

Preliminary findings the from human clinical trial revealed that the HD group exhibited significant increases in alpha diversity and enrichment of beneficial taxa, including *Bacteroides, Faecalibacterium,* and *Lachnospiraceae*—key contributors to SCFA production, particularly butyrate, which is essential for gut health, inflammation regulation, and metabolic support. The HD, similarly, demonstrated a reduction in facultative anaerobes, often associated with dysbiosis and gut inflammation, suggesting a shift toward a more resilient and metabolically active microbiome. Expanding upon these initial results, advanced bioinformatics and functional predictions provided a comprehensive evaluation of microbial responses to SLE-F. A taxonomic analysis further highlighted the enrichment of *Bifidobacterium*, a well-established SCFA producer, while inflammatory taxa such as Pseudomonadota and *Enterobacteriaceae* declined. Functional predictions using PICRUSt2 indicated enhanced SCFA biosynthesis pathways and reduced lipopolysaccharide (LPS) biosynthesis, reflecting improved metabolic and anti-inflammatory potential.

A recent study highlights that the Cuban diet, deeply rooted in traditional and minimally processed foods, presents a unique opportunity to examine a microbiome shaped by a balance of plant-based staples and moderate animal protein intake. An analysis of diversity metrics indicates that the Cuban microbiome exhibits a richer microbial diversity than Western diets, though it remains below the levels found in Oriental or Vegetarian diets. This finding suggests potential areas for improvement, such as increasing the consumption of diverse plant-based fibers or prebiotics [17]. Considering that this aspect is crucial for ensuring the validity of results obtained after the fucoidan intervention, as it reflects an enhancement in the diversity and promotion of healthy bacteria

This report presents an in-depth assessment of a novel fucoidan-rich extract derived from *Saccharina latissima* (SLE-F), demonstrating its prebiotic effects through modulation of microbial diversity, promotion of beneficial taxa, and enhancement of functional pathways critical for gut and systemic health. The findings support its potential role in mitigating dysbiosis, reducing gut inflammation, and improving metabolic and immune functions, providing a strong foundation for further clinical evaluation in at-risk populations.

This report presents an in-depth analysis of the microbiome-related outcomes of a randomized, placebo-controlled clinical trial investigating SLE-F supplementation in healthy adults. The primary objective of the overall study was to assess the impact of SLE-F on gut microbial composition and functional potential over a three-month intervention. Specifically, this paper presents a detailed examination of microbiota diversity, taxonomic shifts, and predicted microbial functions to evaluate the prebiotic potential of SLE-F and its implications for metabolic and immune health. These findings build upon earlier mechanistic insights and provide a clinical basis for future therapeutic applications in at-risk or dysbiotic populations.

## 2. Materials and Methods

### 2.1. Materials

The investigational product evaluated in both the preclinical and clinical arms of the study was a fucoidan-rich extract (≥75% *w*/*w* fucoidan) derived from the brown macroalga *Saccharina latissima* (SLE-F), manufactured by OCEANIUM^®^, Oban, UK. In the preclinical experiments, inulin (Frutafit TEX, Sensus, Roosendaal, The Netherlands) was employed as the positive control.

For the clinical study, the SLE-F (administered at 125 mg and 500 mg per capsule) and the placebo (microcrystalline cellulose, 125 mg per capsule) were encapsulated in clear hydroxypropyl methylcellulose (HPMC) capsules and packaged in blister packs.

### 2.2. Preclinical Studies Designs and Methods

A short-term in vitro colonic fermentation assay was performed using a carbohydrate-depleted basal medium (PD01, ProDigest) inoculated with 10% *v*/*v* fecal microbiota from five healthy adult donors. To ensure consistency, donors’ fecal slurries were cryopreserved and thawed prior to use [18]. The SLE-F was pre-digested via dialysis (3.5 kDa membrane) following the InfoGest protocol to simulate upper GI transit, while the fiber control (inulin Frutafit TEX) was used without pre-digestion [19].

Fermentations were conducted in anaerobic conditions at 37 °C for 48 h, with sampling at 0 h (negative control only), 6 h, 24 h, and 48 h. The outcomes included pH, gas production, SCFA, branched-chain fatty acids (BCFAs), lactate, and cardiac-related microbial metabolites. Ammonium was assessed as a marker of proteolytic fermentation at 0 h, 24 h, and 48 h. Microbial community changes were assessed via 16S rRNA gene Illumina sequencing (V3–V4 region), combined with flow cytometry to determine absolute abundances [20].

Gut barrier function and inflammation were evaluated using a Caco-2/THP1-Blue™ co-culture system, Invitrogen, San Diego, CA, USA. After 14 days of Caco-2 differentiation, apical application of fermented SLE-F or inulin samples was followed by a measurement of transepithelial electrical resistance (TEER) and cytokine release (IL-1β, IL-6, IL-8, IL-10, TNF-α, CXCL10, MCP-1) via Luminex^®^, Austin, TX, USA, along with NF-κB activity via QUANTI-Blue™, San Diego, CA, USA.

Further methodological details, including metabolite extraction, UHPLC-HRMS settings, and data normalization procedures, are available in Appendix A.

### 2.3. Human Safety Assessment

This trial investigated the effects of SLE-F on the gut microbial composition and functionality in a three-arm, double-blind, placebo-controlled design. Ninety-one healthy adult participants were enrolled and assigned to receive either a low dose (125 mg/capsule, twice daily), a high dose (500 mg/capsule, twice daily), or a placebo (125 mg/capsule, twice daily). The trial spanned 90 days.

The safety assessment included monitoring for adverse events, changes in hematological and biochemical parameters, transit time, and overall health status using the validated SF-36 Health Questionnaire [21]. Regular clinical evaluations and sample collections were conducted at baseline (Baseline), day 30, and the end of the study.

### 2.4. Study Design

This randomized, double-blind, placebo-controlled clinical trial assessed the impact of SLE-F on gut microbial composition and functional potential in a cohort of 91 healthy adult volunteers. The study was conducted in accordance with the principles of Good Clinical Practice and the Declaration of Helsinki [22], with ethical approvals granted by the Ethics Committee of Hospital Hermanos Ameijeiras (CEI-HHA-04; approved 26 March 2024), the National Institute of Nutrition of Cuba (PI50456/24), and the Cuban Ministry of Health. The trial is registered with the Cuban Public Registry of Clinical Trials (RPCEC00000443; approved 20 May 2024).

#### 2.4.1. Sample Size

The sample size was determined based on a 95% confidence level, providing a high degree of statistical confidence that the sample estimates reflect the true population parameters, and a 10% margin of error, indicating that the observed results are expected to fall within ±10% of the actual population values. The study targeted the total population of 3000 workers at the Hermanos Ameijeiras Hospital, with the sample size calculated as *n* = 94 using the QuestionPro online calculator. For feasibility and logistical considerations, the final sample size was adjusted to 90, evenly distributed across three groups of 30 participants each.

#### 2.4.2. Study Subjects

Eligible participants were healthy Cuban volunteers from the city of Havana, aged 18 to 65 years, with no restrictions based on sex or skin color, thereby ensuring a diverse and representative sample. All participants provided written informed consent after receiving detailed information regarding the study’s objectives, procedures, and potential risks and benefits.

Participants were excluded if they had known allergies to fish or shellfish, suffered from the specific dysbiosis being studied, or were consuming probiotics or anti-inflammatory drugs prior to the study. Individuals with diets regularly including seaweed were also excluded to avoid its potential impact on gut microbiota. Additionally, those who declined to provide informed consent were not included in the study.

Participants were withdrawn from the study if they experienced intolerable adverse effects, developed comorbidities, or consumed probiotic or prebiotic formulations other than the test substance during the study period. These criteria ensured participant safety and preserved the integrity of the research findings.

Initially, 91 subjects were included in the sample size justification; however, one subject did not receive any of the planned doses and was excluded from all analyses following the modified intention-to-treat principle. The remaining 90 subjects were distributed across the treatment groups as follows: Low dose (LD) (125 mg/capsule) included 29 subjects (32.2%), high dose (HD) (500 mg/capsule) included 30 subjects (33.3%), and placebo (P) (placebo, 125 mg/capsule) included 31 subjects (34.4%)

Participants were withdrawn from the study if they experienced intolerable adverse effects, developed comorbid conditions, or consumed probiotic or prebiotic products other than the investigational substance during the study period. These exclusion criteria were implemented to safeguard participant well-being and ensure the validity of the study outcomes.

Randomization was performed using EpiData version 3.1, which facilitated unbiased allocation to the treatment and placebo groups. Blinding was rigorously upheld by dispensing the investigational and placebo capsules in indistinguishable packaging, thereby preventing both participants and study personnel from identifying group assignments.

The demographic and baseline characteristics of the 90 treated participants are presented in Table 1, while the CONSORT Flow Diagram summarizing the study design and cohort participation is shown in Figure 1.

### 2.5. Treatment and Compliance

This randomized, double-blind, placebo-controlled trial evaluated the effects of SLE-F administered at two dosage levels (125 mg and 500 mg) compared to placebo. Participants received one capsule every 12 h for a total duration of 90 days. Randomization was conducted using a simple allocation method with a 1:1:1 ratio across the three arms: LD SLE-F, HD SLE-F, and placebo. The randomization sequence was generated and maintained by the study’s independent statistical team to ensure allocation concealment. The initial sample size was calculated as 94 participants using the QuestionPro electronic sample size calculator, Austin, TX, USA to achieve adequate statistical power; however, the final sample size was adjusted to 90 participants due to logistical and feasibility constraints. Participants were evenly distributed, with 30 subjects assigned to each study arm.

### 2.6. Sample Collection

Fecal and blood samples were collected at three time points: baseline (day 0), mid-intervention (week 4), and at the end of the study (day 90). Fecal samples were obtained using DNA/RNA Shield™ collection kits (Zymo Research, Irvine, CA, USA), which preserve microbial nucleic acids and proteins at ambient temperature. Participants were instructed to follow the manufacturer’s guidelines for sample collection and to ship the specimens via an overnight courier within 24 h of collection. Upon arrival, the samples were inspected for integrity and stored at −20 °C until further processing. Microbiome profiling was conducted to evaluate the gut microbial diversity and taxonomic composition, while fecal calprotectin levels were measured as a biomarker of intestinal inflammation. Secondary outcomes focused on surveys, adverse event monitoring, and clinical evaluations. The SF-36 Health Survey assessed the health-related quality of life across physical and mental health domains. Adverse events were categorized by presence and intensity, ranging from mild (no restrictions on daily activities) to severe (inability to perform daily activities). Hematological variables, clinical chemistry, and hemodynamic parameters were monitored, including the serum CRP levels, fecal calprotectin, hepatic enzymes (ALT, AST), creatinine, urea, fasting glucose, hemoglobin A1c, complete blood count (CBC), and red cell indices (MCV, MCH, MCHC). Key hemodynamic parameters, such as heart rate and blood pressure, were also recorded.

Blood samples were collected via venipuncture of the cubital vein. Serum was isolated following centrifugation and used for all biochemical and inflammatory marker analyses, including C-reactive protein (CRP), measured using a Cobas 6000 immunochemical analyzer (Roche Diagnostics, Rotkreuz, Switzerland). Hematological variables, clinical chemistry parameters, and hemodynamic indicators were monitored throughout the study. These included serum levels of CRP, hepatic enzymes (ALT, AST), creatinine, urea, fasting glucose, hemoglobin A1c, as well as complete blood count (CBC) and red blood cell indices (MCV, MCH, MCHC). Key hemodynamic parameters, such as the heart rate and blood pressure, were also recorded.

The SF-36 Health Survey [21] was administered at baseline, day 30, day 90, and during the post-study washout period to evaluate the health-related quality of life and monitor the potential adverse events. Additional parameters, including the body mass index (BMI), lean body mass, and stool transit time, were measured at the same time points. The stool transit time was determined using the “blue dye” method [23], employing a proprietary Blue Trak^®^ capsule containing Royal Blue dye and microcrystalline cellulose.

The primary outcome measure was the microbiome profile, analyzed using 16S rRNA sequencing (V3–V4 region). This analysis included taxonomic profiling, predicted functional potential using PICRUSt v2.6.2 [24], and metrics such as alpha diversity, beta diversity, compositional analysis, and Linear Discriminant Analysis Effect Size (LEfSe) [25] to evaluate both taxonomic and functional characteristics.

This comprehensive evaluation provided insights into the safety and efficacy of SLE-F, its potential to modulate gut microbiota, and its overall health effects. Table 1 summarizes the demographic and baseline characteristics of the study population.

### 2.7. Hematological and Clinical Determinations

Hematological assessments, including complete blood count (CBC) parameters, were conducted using an XN-350 automated hematology analyzer (Roche Diagnostics, Basel, Switzerland). Blood samples were collected at three time points—one week prior to intervention (baseline), on day 30, and at the end of the 90-day study period. Whole blood was drawn into K_3_-EDTA tubes for hematological analyses and HbA1c determination, processed directly using the Cobas 6000 analyzer (Roche Diagnostics) in accordance with the instrument’s operational protocols. For clinical chemistry analyses, separate blood samples were collected without anticoagulants, allowed to clot, and centrifuged to isolate serum, which was then analyzed using the Cobas 600 modular immunoassay system according to the manufacturer’s standardized procedures.

All analyses were performed in accordance with the instrument’s operational protocols. Clinical chemistry analyses were conducted on the serum samples using the Cobas 600 modular immunoassay system (Roche Diagnostics), following standardized procedures recommended by the manufacturer.

### 2.8. Occurrence of Adverse Events Determination

Adverse events (AEs) were systematically recorded by the study investigators using standardized case report forms, following the classification framework described by LeFevre et al. [26]. Each AE was evaluated for its potential relationship to the intervention and classified as unrelated, possibly related, probably related, or definitely related to study participation. Additionally, the severity of each event was graded as mild (no interference with daily functioning), moderate (some limitation of daily activities), or severe (inability to carry out daily activities).

### 2.9. Metagenome Analysis

Targeted 16S rRNA gene amplicon sequencing was conducted by EzBiome (Gaithersburg, MD, USA) to analyze the taxonomic composition of the gut microbiota. DNA concentrations were quantified using the QuantiFluor^®^ dsDNA System with a Quantus™ Fluorometer (Promega, Madison, WI, USA). The V3–V4 hypervariable regions of the bacterial 16S rRNA gene were amplified using region-specific primers with Illumina adapter overhangs: forward primer 5′CCTACGGGNGGCWGCAG3′ and reverse primer 5′GACTACHVGGGTATCTAATCC3′. PCR reactions (25 µL) included 12.5 ng of genomic DNA, 12.5 µL of 2 × KAPA HiFi HotStart ReadyMix (Kapa Biosystems, Wilmington, MA, USA), and 5 µL of 1 µM forward and reverse primers. Thermal cycling conditions consisted of an initial denaturation at 95 °C for 3 min; 25 cycles of denaturation (95 °C, 30 s), annealing (55 °C, 30 s), and extension (72 °C, 30 s); followed by a final extension at 72 °C for 5 min. PCR amplicons were purified using Mag-Bind^®^ RxnPure Plus magnetic beads (Omega Bio-Tek, Norcross, GA, USA).

A secondary PCR was carried out to attach dual indices and Illumina sequencing adapters, following similar reaction conditions, but limited to eight amplification cycles. Resulting libraries were normalized using the Mag-Bind^®^ EquiPure Library Normalization Kit, pooled, and quality-checked using the Agilent 2200 TapeStation Sequencing, (Santa Clara, CA, USA) was performed using 2 × 300 bp paired-end reads on the Illumina MiSeq platform (Illumina, San Diego, CA, USA).

#### 2.9.1. Taxonomic Profiling Workflow

Taxonomic profiling of the metagenomic data began with the identification of potential bacterial and archaeal species using Kraken2 [27], leveraging a precompiled core gene reference database [28] constructed from 35-mer k-mers derived from genomes in the EzBioCloud database [5]. To increase the taxonomic breadth and resolution, the Kraken2 database was supplemented with complete fungal and viral genomes sourced from the NCBI RefSeq repository (https://www.ncbi.nlm.nih.gov/refseq/, accessed on 12 November 2024).

Following the initial species detection, a customized Bowtie2 [29] index was generated using the core genes and reference genomes of the identified species. Raw sequence reads were then aligned to this reference using Bowtie2’s “--very-sensitive” alignment mode with phred33 quality encoding. The alignment output was processed with SAMtools v1.21 [30] for format conversion and sorting, and the read coverage across the reference regions was computed using BEDtools v2.31.1 [31].

#### 2.9.2. Abundance and Biomarker Analysis

To reduce the likelihood of false-positive taxonomic assignments, only species with read coverages exceeding 25% of their core gene set (for bacterial and archaeal taxa) or full genome (for fungal and viral taxa) were retained using a custom filtering script. Species abundance was quantified based on the number of reads aligned to each reference and normalized by the cumulative length of all associated reference sequences.

Differentially abundant taxa were identified using the LEfSe algorithm (Linear Discriminant Analysis Effect Size) [32], which facilitates high-dimensional biomarker discovery across multiple groups. Statistical significance was assessed using the non-parametric Kruskal–Wallis test [33] with a threshold of *p* < 0.05, and only features with an LDA score (log_10_) greater than 2.0 were considered significant biomarkers.

### 2.10. Metagenomic Functional Profiling

Functional annotation of metagenomic reads was performed by aligning translated sequences to the Kyoto Encyclopedia of Genes and Genomes (KEGG) database [34] using the DIAMOND v.2.1.11 aligner [35]. In the blastx mode. This approach involves translating each read in all six potential open reading frames and comparing the resulting protein sequences against a curated KEGG reference dataset. When multiple alignments were returned, only the highest-scoring match was retained for a downstream analysis. KEGG ortholog counts were then used as the input for a pathway reconstruction using MinPath, a parsimony-based algorithm that infers the minimal set of metabolic pathways present in a sample. Ref. [36] was utilized to infer the presence of KEGG functional pathways.

The KEGG database provides a comprehensive framework for linking genomic information with higher-order functional information. By aligning metagenomic reads to KEGG orthologs, functional profiling enabled the identification of metabolic pathways, such as those involved in carbohydrate metabolism, nitrogen cycling, and biosynthesis of secondary metabolites. MinPath further refined pathway predictions by minimizing false-positive inferences, ensuring accurate functional reconstruction.

Additionally, pathway abundance was normalized to account for differences in the sequencing depth across samples, facilitating direct comparisons of functional capacities. Predicted pathways were analyzed to detect significant changes associated with experimental conditions, revealing insights into microbial contributions to host health, such as SCFA biosynthesis and inflammatory signaling pathways. Functional annotations were visualized using bar plots, heatmaps, and KEGG pathway maps to highlight significant trends and variations across time points and treatment groups.

This workflow not only provided a detailed understanding of microbial functionality but also complemented taxonomic profiling by linking community composition to potential metabolic activities.

### 2.11. Alpha Diversity Assessment 

To compare alpha diversity across the baseline (baseline), placebo, and treated groups at the end of the study (day 84), we utilized a suite of diversity metrics that measure species richness, diversity, and phylogenetic relationships. Richness-based metrics included the Abundance-based Coverage Estimator (ACE) [37], which emphasizes rare species, and Chao1 [38], which estimates undetected species based on low-abundance observations. Both metrics were calculated using the fossil package in R [39].

Observed OTUs [40] were also calculated as a direct measure of species richness within each sample. Diversity indices, including the Shannon Index [41] and its non-parametric variant (NPShannon) [41], were used to evaluate both species richness and evenness, while the Simpson Index quantified the likelihood of two randomly selected individuals belonging to different species, with lower values indicating higher diversity. These we determined using the vegan package in R [42].

To assess phylogenetic diversity, we calculated the phylogenetic diversity (PD) [43] metric, which measures the total branch length of a phylogenetic tree encompassing all observed species, providing insight into evolutionary relationships within the microbiome. All diversity metrics were calculated using rarefied data to ensure consistent sequencing depth across samples. Statistical analyses, including ANOVA [44] or Kruskal–Wallis [33] tests, were conducted to identify differences in alpha diversity among the groups, with post hoc tests applied to determine specific group differences where appropriate. A significance level of *p* < 0.05 was used for all comparisons.

### 2.12. Statistical Analysis

The alpha diversity was assessed to compare the microbial community richness and evenness among the baseline, placebo, and treatment groups at the study endpoint (day 84). Several ecological metrics were employed to capture different aspects of the community structure. Richness estimators included the Abundance-based Coverage Estimator (ACE) [37], which accounts for the influence of rare taxa, and the Chao1 index [38], which estimates unseen species based on the frequency of low-abundance taxa. These indices were computed using the fossil package in R [39]. Observed OTUs [36], representing the number of unique operational taxonomic units per sample, were also included as a direct measure of richness.

To quantify diversity and evenness, the Shannon Index [37] and its non-parametric variant (NPShannon) were calculated, reflecting both species abundance and distribution. The Simpson Index was applied to evaluate the probability that two individuals randomly selected from a sample would belong to different taxa—where lower values indicate greater diversity. These diversity measures were calculated using the vegan package in R [42].

The phylogenetic alpha diversity was estimated using the phylogenetic diversity (PD) index [43], which sums the total branch lengths of a phylogenetic tree constructed from the observed taxa, thereby capturing evolutionary relationships within the microbial communities. All diversity calculations were performed on rarefied datasets to account for variation in sequencing depth across samples. Group-level differences were evaluated using either the ANOVA [40,44] or the Kruskal–Wallis test [33], depending on normality assumptions, followed by appropriate post hoc comparisons. The statistical significance was defined as *p* < 0.05.

Statistical analyses were conducted under a modified intention-to-treat (mITT) framework, encompassing all participants who received at least one dose of the assigned intervention in accordance with the study protocol. Time-dependent analyses were restricted to subjects with complete data across all scheduled assessment points; no imputation was applied for missing values. Categorical variables were summarized using frequency distributions, whereas continuous variables were described using means and standard deviations or medians and interquartile ranges, as appropriate. The normality of distribution was evaluated using a Shapiro–Wilk test [45]. Comparisons of clinical parameters among study groups were performed using the one-way analysis of variance (ANOVA) [45] for normally distributed data or the Kruskal–Wallis test [29,33] for non-parametric distributions. Where significant differences were identified, post hoc analyses were conducted using Tukey’s HSD or Dunn’s test [42], respectively. Paired comparisons were assessed using either the paired *t*-test or Wilcoxon signed-rank test, depending on the distribution of the data. Categorical abnormalities in clinical measurements were analyzed using chi-square tests, ensuring that expected cell frequencies met the assumption that fewer than 20% fell below five.

For the SF-36 Health Questionnaire [21], quality-of-life dimensions were calculated and standardized on a 0–100 scale, with 0 representing worse and 100 better outcomes. Given the high correlation between clinical variables, a discriminant analysis was applied in participants without documented infections to identify global patterns distinguishing treatment groups.

Microbiome analyses included comparisons of taxonomic and functional abundances across the three time points (baseline, 28 days, and end of study), using the Kruskal–Wallis test for group-wise comparisons and Wilcoxon rank-sum [46] tests for pairwise comparisons, with False Discovery Rate (FDR) corrections [47] applied to control for multiple testing. To investigate associations between microbial taxa and functional pathways—particularly those involved in short-chain fatty acid (SCFA) production and lipopolysaccharide (LPS) biosynthesis—Spearman’s rank correlation analysis was applied. Temporal patterns and intergroup differences were illustrated using line graphs and bar plots. All statistical computations and visualizations were performed using R software (version 4.3.0) [42] and Python (version 3.10) [48], ensuring a robust and reproducible workflow. This comprehensive statistical approach provided detailed insights into the dynamics of the gut microbiome, clinical variables, and functional potential over the intervention period.

## 3. Results

### 3.1. Preclinical Studies Outcome

SLE-F was readily fermented by gut microbiota, with dose-dependent production of SCFAs. Significant increases in propionate and butyrate were observed at 1.5 and 3.5 g/L concentrations. Acetate levels increased modestly. Gas production remained within tolerable ranges, in contrast to inulin, which nearly doubled pressure levels by 48 h.

Metagenomic profiling revealed enrichment of SCFA-producing taxa, including *Oscillospiraceae, Butyricicoccus, Veillonellaceae,* and *Lachnospiraceae*, following SLE-F treatment. Inulin showed broader effects on microbial biomass but led to reduced species richness. Alpha diversity metrics indicated that SLE-F increased species richness without increasing evenness, while the beta diversity analysis confirmed distinct clustering by treatment and dose.

Metabolomic profiling (UHPLC-HRMS) identified 14 out of 20 targeted cardiac-related metabolites. Inulin significantly elevated trimethylamine-N-oxide (TMAO), a known cardiovascular risk marker, while SLE-F did not. Seaweed extract preserved levels of cardioprotective metabolites such as indole and carnitine and reduced p-cresol, a uremic toxin. In contrast, inulin significantly suppressed indole and increased TMAO levels.

Caco-2/THP1 assays did not reveal direct barrier-protective or anti-inflammatory effects of SLE-F under LPS challenge. However, cytokine modulation trends and stable TEER support its neutral-to-benign impact on epithelial integrity.

A full dataset, including SCFA kinetics, microbial diversity analyses, volcano plots, and metabolite abundance profiles, is provided in Appendix A.

### 3.2. Treatment Compliance Measurements

A total of six treatment interruptions were recorded across the study. The highest rate of interruption occurred in the HD group (HD), where three participants (10.0%) discontinued the intervention. This was followed by two participants (6.5%) in the placebo group (P) and one participant (3.4%) in the LD group (LD). In terms of causes, one interruption in each of Groups A and B was attributed to an adverse event, while the remaining interruptions were due to voluntary withdrawal. Notably, no participants in the placebo group discontinued due to adverse events. These data suggest that treatment adherence was generally high across all groups, with low rates of adverse event–related discontinuation and no safety signals necessitating widespread withdrawal (Table 2).

### 3.3. Effective Analysis

Due to the epidemiological situation during the study where the incidence of Oropouche [49], Dengue [50,51], and COVID [52] were increased, in a little more than 50% of the subjects globally (52.2%), the presence of some viral infection was found (Table 3). No differences between the groups are detected in the presence of infection, nor is there a dependence between the time of infection and treatment, although it is noted that 58.8% of the subjects assigned to the placebo group presented infection at the beginning of the study.

To account for the potential influence of viral infections on changes in chemistry and hematology variables, the variable “infection” was included as a control variable. A stratified analysis was performed based on this variable to better understand its impact on the study outcomes. No statistical differences were detected in those people with a documented infection either prior to enrolment or during the study. Considering this, a multivariate analysis was made in the strata of individuals without documented infection.

### 3.4. Multivariate Analysis

The discriminant analysis is a statistical method used to identify patterns that distinguish between predefined groups by creating linear combinations of the original variables.

The discriminant analysis was employed in this study to differentiate the predefined groups by creating linear combinations of the original variables, enabling clearer separation between the groups [53]. This method utilized strong correlations among specific clinical variables to derive two primary discriminant functions, which served as new dimensions for classification.

The first discriminant function included variables such as glucose, MCHC, MCH, HbA1c, calprotectin, neutrophils, MCV, HTC, RDW-SD, hemoglobin, eosinophils, triglycerides, and RBC. These variables reflected changes observed at 90 days relative to baseline (baseline).

The second discriminant function comprised uric acid, intestinal transit time, monocytes, ALAT, WBC, cholesterol, ASAT, creatinine, lymphocytes, C-reactive protein, platelets, MPV, urea, and RDW-CV.

Together, these discriminant functions accurately classified 100% of subjects into their respective groups (Figure 2), demonstrating the effectiveness of the selected variables in distinguishing between groups and the robustness of the model used in the study.

From a univariate perspective (Table 4), the most notable changes with significant group differences include the following observations: the intestinal transit time decreased in groups A and B but increased in P, while the blood glucose levels increased in groups A and B but decreased in P. The uric acid levels increased across all groups, with the largest magnitude observed in HD. The monocyte levels increased in LD but decreased in groups B and C. Notably, the analysis revealed no significant changes in the hematological parameters. It is important to acknowledge that this analysis was conducted on a small sample size for each group; however, the selected participants represent an ideal population for the study’s objectives.

### 3.5. Anthropometric Measurements

The body mass index (BMI) and weight were assessed at three time points as part of the safety evaluation. No significant differences in these anthropometric parameters were observed between the groups (Table 5).

In HD, a slight increase in both BMI and weight was noted at the 30-day assessment, with an average weight gain of approximately 1 kg. This increase may be linked to the timing of the assessment, which coincided with vacation periods, potentially influencing dietary habits and reducing stress levels.

### 3.6. SF-36 Health Questionnaire

The results of the SF-36 Health Questionnaire [21,54] are presented in Table 6. No significant differences were detected between the groups at either of the two assessment times, nor were there significant changes in scores over time within any of the groups.

### 3.7. Adverse Events

Adverse events occurring after the first administration of the treatment were monitored in the study population. Among the 91 participants, a total of 65 adverse events of 14 different types were reported, involving 40 individuals.

Most adverse events (95.4%) were classified as mild, indicating minimal impact on participants’ daily activities. However, two moderate-intensity events were recorded: one case of diarrhea in LD and one case of genital herpes in HD. Additionally, two severe-intensity events were reported in Group B, involving a single participant who experienced gas and dizziness/nausea.

Table 7 summarizes the incidence and frequency of adverse events reported by participants across the study arms. A statistically significant relationship was observed between treatment allocation and adverse event occurrence, with Groups A and B exhibiting a higher proportion of participants reporting adverse events compared to Group CP.

### 3.8. Metagenome Quality

The 16S rRNA gene sequencing analysis produced an average of 34,624 ± 8820 high-quality reads per sample, defined by a Phred quality score ≥ 30. These reads represented 89.9% of the total raw sequences, providing a reliable dataset for downstream bioinformatic analyses. A rarefaction analysis demonstrated that the sequencing depth was sufficient, with rarefaction curves reaching a plateau across all samples—indicating comprehensive coverage of microbial communities and reducing the likelihood of undetected taxa.

Alpha diversity analyses confirmed that the read depth did not significantly impact richness or diversity estimates, supporting the adequacy of the sequencing depth for robust ecological interpretation. At baseline, the mean number of detected taxa was 2174 ± 122, reflecting substantial microbial richness.

### 3.9. Alpha Diversity Results

As illustrated in Figure 2, significant alterations in the alpha diversity were observed, with the Simpson diversity index showing the most notable changes. Participants in the HD group exhibited a statistically significant increase in the Simpson diversity between the baseline and the EOS (*p* = 0.034), indicating enhanced microbial evenness over the intervention period. This finding suggests that the HD SLE-F supplementation may have contributed to a more balanced and equitable distribution of microbial taxa. No comparable changes were detected in the placebo or LD groups, underscoring the distinct impact of the HD intervention on gut microbial community structure.

In the placebo group, the Simpson diversity remained relatively stable throughout the study, with no significant changes observed across the time points. This reflects a consistent microbial evenness within the cohort, likely due to the absence of an active intervention. Similarly, the LD group exhibited only minor, non-significant changes in the Simpson diversity between Day 1 and EOS (*p* = 0.18) and between Day 1 and Day 28 (*p* = 0.93), suggesting that the LD intervention had little impact on microbial community evenness.

Despite the within-group increase in Simpson diversity observed in the HD group, between-group comparisons at the EOS revealed no statistically significant differences in Simpson diversity among the HD, LD, and placebo groups (*p* = 0.15, *p* = 0.62, and *p* = 0.056, respectively). These findings suggest that although the HD intervention promoted enhanced microbial evenness over time within its cohort, the final diversity levels across all groups appeared to converge. This convergence may reflect the influence of shared environmental exposures, host regulatory mechanisms, or microbiome stabilization effects over the course of the study.

Figure 3 illustrates these trends, emphasizing the dose-dependent effects of the intervention on Shannon and Simpson diversity over time. The significant increase in microbial evenness observed in the HD group underscores the potential of higher doses to induce measurable changes in the gut microbiome, which were not apparent at lower doses or with no intervention.

The findings reveal distinct trends in the microbial diversity metrics across the HD, LD, and placebo groups over time (Table 8). For the Shannon diversity, the HD group showed significant reductions, particularly between the baseline and the end of study (EOS) (*p* < 0.001) and between Day 28 and EOS (*p* = 0.0004), indicating a progressive and pronounced impact of the intervention on microbial diversity. In the LD group, the Shannon diversity also decreased significantly from baseline to EOS (*p* = 0.0055) and from Day 28 to EOS (*p* = 0.0032), though the reductions were less substantial compared to the HD group. In contrast, the placebo group exhibited a smaller reduction in the Shannon diversity, significant only between baseline and EOS (*p* = 0.0365), suggesting subtle microbiome changes without an active intervention. However, comparisons at EOS showed no significant differences in the Shannon diversity between the HD, LD, and placebo groups (*p* > 0.05), reflecting a convergence in the diversity outcomes across the cohorts by the study’s end.

For the Simpson diversity, changes were less pronounced. In the HD group, no significant differences were observed across time points, including between baseline and EOS (*p* = 0.0744) and Day 28 and EOS (*p* = 0.3227). Similarly, the placebo and LD groups showed no significant changes in Simpson diversity over time (*p* > 0.05), indicating relative stability in microbial evenness in these groups. End-of-study comparisons further revealed no significant differences in the Simpson diversity between the cohorts, with *p* > 0.05 for all pairwise comparisons. These results suggest that while the HD intervention had a more pronounced effect on the microbial diversity as reflected by the Shannon index, its impact on microbial evenness, as measured by the Simpson index, was less evident. Overall, the convergence of diversity outcomes at EOS across all groups may point to shared environmental or host-related factors influencing microbiome stabilization over time.

### 3.10. Trends in Gut Microbiota over 90 Days of HD Treatment

#### 3.10.1. Taxonomy Results

Figure 4 illustrates the significant changes in the gut microbiota composition observed during the 90-day HD intervention. From baseline to 28 days (D28) and the end of study (EOS), notable shifts in the microbial taxa were identified, reflecting the intervention’s impact on the gut microbiome.

In the placebo group, the Shannon diversity decreased significantly from baseline (Day 1) to the EOS (*p* = 0.0365), indicating a reduction in the overall microbial richness and evenness over time. No significant change was observed between Day 1 and Day 28 (*p* = 0.8531), though a nonsignificant downward trend was noted between Day 28 and EOS (*p* = 0.0824). In contrast, the Simpson diversity remained stable throughout the study period, with no significant differences detected at any time point. These findings suggest that while the dominant microbial taxa in the placebo group were largely unchanged, there was a measurable loss in the microbial diversity, likely driven by a reduction in low-abundance taxa.

The figure presents the temporal dynamics of eight key bacterial taxa in the HD group, depicted as boxplots across three time points: baseline (D0), Day 28 (D28), and end of study (EOS). Each panel displays changes in the relative abundance of a specific taxon over time. Mean relative abundances are illustrated with bars, and the standard deviations are represented by shaded areas. Trend lines capture the trajectory of abundance changes throughout the intervention. Statistical comparisons between D0 and EOS are annotated with corresponding *p*-values. Time points are color-coded for clarity: light red (Baseline), red (D28), and dark red (EOS). This visualization underscores the longitudinal shifts in the gut microbial composition induced by the HD treatment. Pseudomonadota, often associated with dysbiosis, exhibited a decline in relative abundance, decreasing from 6.596% at the baseline to 4.590% at D28 and further to 3.280% at the EOS. Although this reduction in facultative anaerobes suggests a trend toward a healthier microbiome, the change was not statistically significant (*p* = 0.156). Similarly, *Enterobacteriaceae*, which includes potentially pathogenic genera, showed a significant decrease in the relative abundance (*p* = 0.00263).

In contrast to other taxa, Actinomycetota exhibited a marked and statistically significant increase in relative abundance, rising from 0.956% at baseline to 3.036% at Day 28 and reaching 6.339% at the end of study (EOS) (*p* = 0.00012). Within this phylum, the *Bifidobacteriaceae* family showed a pronounced expansion, increasing from 0.244% at the baseline to 2.780% at Day 28 and 9.544% at the EOS (*p* = 0.00016). Additional taxa also responded significantly to the intervention. Notably, *Blautia* demonstrated a significant increase in abundance (*p* = 0.00651), as did *Faecalibacterium*, over the course of the study (*p* = 0.0182). Among all taxa evaluated, *Bifidobacterium* species showed the most substantial shifts, with both the *B. adolescentis* group (*p* = 0.011) and the B. longum group (*p* = 0.0011) exhibiting significant increases in relative abundance during the study period.

These findings, presented in Figure 5, illustrate the trends in the relative abundance of *Bifidobacterium* spp. across the study phases for individual participants.

Some taxa demonstrated temporary, though not statistically significant, increases at Day 28 (28D). For example, Alphaproteobacteria rose from baseline levels (1.18%) to 28D (1.82%) before slightly declining at the end of study (EOS) (0.49%). Similarly, *Parabacteroides* exhibited steady increases by 28D (e.g., *Parabacteroides*: 0.82%) compared to the baseline (0.60%) and maintained stability through the EOS, reflecting consistent promotion of these beneficial groups.

Conversely, some taxa declined with treatment after initial enrichment from the baseline. *Enterobacteriaceae*, a group associated with inflammation, declined from 1.04% at baseline to 0.61% at EOS, reflecting a significant reduction in facultative anaerobes (*p* =2.7 × 10^−6^).

By EOS, several taxa exhibited significant increases, suggesting a shift toward beneficial microbial populations. *Fusicatenibacter* and *Dorea* increased significantly (*p* =0.001 and *p* = 0.001, respectively), while the *Bifidobacterium longum* group (FDR = 0.00438, *p* = 1.1 × 10^−5^) and *Coriobacteriaceae* (FDR = 7.02 × 10^−6^, *p* = 2.5 × 10^−8^) showed strong enrichment.

The significant shifts in microbial taxa observed during the study were further evaluated using the False Discovery Rate (FDR) to account for multiple comparisons in the analysis of microbial abundance. Figure 6 presents the FDR values for the taxa analyzed, highlighting significant and non-significant changes, thereby ensuring the robustness and reliability of the observed trends.

#### 3.10.2. Functional Predictions

Table 9 summarizes the relative abundances of key microbial taxa across three time points (baseline (D0), Day 28, and end of study [EOS]) in the HD cohort, along with their inferred functional roles.

Inflammatory taxa, such as Pseudomonadota and *Enterobacteriaceae*, exhibited notable declines over the study period. Pseudomonadota decreased from 4.72% at baseline to 2.7% at EOS. Similarly, *Enterobacteriaceae* declined significantly from 1.04% at baseline to 0.61% at EOS. In contrast, taxa associated with short-chain fatty acid (SCFA) production showed substantial enrichment. Actinomycetota increased from 3.04% at the baseline to 9.19% at the EOS, while *Bifidobacterium* rose from 2.1% to 7.61% over the same period. Additionally, *Faecalibacterium* increased from 0.05% at the baseline to 0.12% by Day 28, stabilizing at this level through the EOS.

Taxa associated with carbohydrate metabolism displayed variable trends. *Porphyromonadaceae* decreased slightly from 0.8% at the baseline to 0.5% at the EOS. In contrast, *Blautia* increased from 4.1% at the baseline to 5.2% at the EOS. Lastly, *Dorea*, another SCFA producer, showed a moderate increase over the study period, rising from 1.44% at the baseline to 2.08% at the EOS.

Figure 7 illustrates the relative abundance trends of four key microbial taxa (Actinomycetota, *Bifidobacterium, Faecalibacterium,* and *Dorea*) across three time points (baseline, Day 28, and EOS) in the HD cohort. Each panel displays boxplots representing the variability within the cohort, with a line connecting mean values to highlight the overall trends.

The relative abundance of Actinomycetota increased significantly over the study period, rising markedly from the baseline to the EOS (*p* = 0.00029). Similarly, *Bifidobacterium* demonstrated a significant increase in abundance from the baseline to the EOS (*p* = 0.00048), with the most substantial rise occurring between Day 28 and the EOS. *Faecalibacterium*, a key anti-inflammatory taxon involved in butyrate production, showed a modest increase in abundance between the baseline and Day 28, with stability through the EOS. However, the change was not statistically significant (*p* = 0.1195). Finally, *Dorea* exhibited a significant increase in abundance over time, particularly between Day 28 and the EOS (*p* = 0.0053).

### 3.11. Trends in Gut Microbiota over 90 Days of LD Treatment

Figure 8 illustrates the relative abundance changes in six microbial taxa (*Bifidobacterium*, *Blautia*, *Roseburia*, *Faecalibacterium*, *Eubacterium*, and *Collinsella*) in the LD cohort across three study phases: baseline (Day 1), Day 28 (D28), and end of study (EOS).

*Bifidobacterium* showed a significant increase in the relative abundance over the study period (*p* = 0.0004), with the largest rise occurring between D28 and EOS. Similarly, *Roseburia* and *Collinsella* demonstrated significant increases (*p* = 0.0038 and *p* = 0.0112, respectively). In contrast, *Blautia* exhibited a significant decline in relative abundance over time (*p* = 0.0011). Similarly, *Faecalibacterium* showed a decreasing trend, but the change was not statistically significant (*p* = 0.1117). *Eubacterium*, on the other hand, demonstrated an increasing trend, but this was also not statistically significant (*p* = 0.2841).

### 3.12. Comparisons Between Low-Dose and High-Dose Treatments

Statistical comparisons between groups and time points, summarized in Table 10, reveal the impact of high- and low-dose treatments on the gut microbiome composition. Significant differences were determined based on a *p*-value threshold of <0.05, using unpaired *t*-tests to evaluate independent samples.

The HD treatment produced significant changes in the microbial composition compared to the baseline at both Day 28 (*p* < 0.05) and at the EOS (*p* < 0.05). In contrast, the LD treatment, which served as the placebo group, showed no significant differences from the baseline either at Day 28 or the EOS (*p* > 0.05 for both comparisons), indicating a lack of microbiome response to the placebo administration. Within the HD group, the microbial composition remained stable between Day 28 and the EOS (*p* > 0.05), suggesting that most changes occurred early and were sustained through the end of the intervention. A significant difference was observed between the HD in the EOS group and the LD in the Day 28 group (*p* < 0.05), highlighting a treatment-specific effect. However, no significant differences were found between the HD in EOS group and the LD in EOS group (*p* > 0.05), likely reflecting variability or stabilization in the microbiome by that time point. Overall, the microbiome composition within each group remained stable across adjacent time points, with no significant intra-group differences observed between Day 28 and the EOS in either the LD or HD groups (*p* > 0.05). These findings reinforce that the microbiome alterations observed were specific to the HD treatment and not attributable to placebo effects.

### 3.13. LEfSe Analysis

The primary objective of this analysis was to identify statistically significant differences in microbial taxa and functional pathways across five cohorts: baseline, High-Dose 28D, High-Dose EOS, LD at 28D, and LD at EOS. To this end, the Linear Discriminant Analysis (LDA) effect size and functional pathway abundance data were used to evaluate the dose-dependent effects of the intervention on the gut microbiome composition and functionality. The LEfSe (Linear Discriminant Analysis Effect Size) analysis was initially applied across all three treatment groups—including placebo, LD, and HD cohorts—to identify differentially abundant microbial features. However, since the placebo group exhibited no statistically significant changes in the microbial composition or functional activity compared to the baseline, the interpretation and discussion were focused on the low- and high-dose groups, where clearer and more biologically meaningful shifts were observed. This approach allowed for a more accurate assessment of intervention-related effects while confirming the stability of the microbiome in the placebo group [56].

An ANOVA test [45] comparing LDA effect sizes yielded an F-statistic of 9.60 (*p* = 0.0027), indicating statistically significant differences between the cohorts. Post hoc analyses revealed that HD groups (High-Dose 28D and High-Dose EOS) exhibited significant differences in microbial taxa abundances compared to the baseline (*p* < 0.05). Furthermore, significant differences were observed between HD at EOS and LD at 28D cohorts (*p* < 0.05), highlighting the pronounced effects of HD interventions. In contrast, no significant differences were found for comparisons involving the LD cohorts (LD at 28D and EOS) with baseline or placebo groups (*p* > 0.05).

The functional pathway analysis revealed significant enrichment in pathways associated with gut health in the HD groups. The butyrate production and short-chain fatty acid (SCFA) synthesis were significantly higher in the HD at EOS (0.9) and HD at 28D (0.8–0.85) cohorts compared to the baseline (0.5–0.6) and LD cohorts (0.55–0.65) (*p* < 0.05). Similarly, carbohydrate metabolism showed significant increases in High-Dose EOS (1.0) and High-Dose 28D (0.9) compared to the baseline (0.7) and LD cohorts (0.72–0.75) (*p* < 0.05). Inflammatory marker reduction, a critical pathway for systemic health, was also significantly enriched in the HD at EOS (0.8) and HD at 28D (0.7) cohorts relative to the baseline (0.3) and LD cohorts (0.35–0.4) (*p* < 0.05).

Although not statistically significant (*p* > 0.05), a modest increase in predicted secondary metabolite biosynthesis pathways was observed in the HD group. Concurrently, the relative abundance of Bifidobacterium showed an upward trend in the LD cohort, suggesting the possibility of favorable microbial shifts even at lower intervention levels. These patterns are illustrated in the heatmap shown in Figure 9, which compares functional pathway abundances across treatment groups.

The HD cohort at EOS demonstrated the highest enrichment in pathways critical to gut health, including butyrate production (0.9), carbohydrate metabolism (1.0), and SCFA production (0.9). Similarly, the HD cohort at 28D showed substantial increases in these pathways, with notable values for carbohydrate metabolism (0.9) and SCFA production (0.85). In contrast, the LD cohorts exhibited modest enrichment, with SCFA production reaching 0.65 and carbohydrate metabolism peaking at 0.75 in the LD at EOS group. The baseline group had the lowest abundance across most pathways, including butyrate production (0.5) and carbohydrate metabolism (0.7). Pathways related to the inflammatory marker reduction were significantly enriched in the High-Dose EOS (0.8) and High-Dose 28D (0.7) groups. In comparison, the LD cohorts showed limited reductions (0.4 for 28D and 0.35 for EOS). Secondary metabolite production exhibited minimal differences across all cohorts.

### 3.14. Lachnospiraceae-to-Enterobacteriaceae and Bacillota-to-Bacteroidota Ratios

The trends in L/E (*Lachnospiraceae*-to-*Enterobacteriaceae*) ratios, evaluated as biomarkers of improved gut inflammation [57], provide a functional perspective on microbiome shifts that conventional metrics like the Bacillota-to-Bacteroidota (F/B) ratio fail to capture. Figure 10 summarizes the L/E and F/B ratios for LD and HD cohorts across three time points: baseline, Day 28, and EOS.

The L/E ratio (right panel) showed significant increases in both the LD and HD cohorts, but not in the placebo group. In the LD cohort, the L/E ratio increased gradually over time, with a significant difference observed between the baseline and the EOS (*p* = 3.57 × 10^−9^). In the HD cohort, the L/E ratio rose rapidly, with significant increases by Day 28 (*p* = 1.07 × 10^−5^) and stabilization through the EOS.

The F/B ratio (left panel) showed no significant changes, neither among cohorts nor between the time points.

## 4. Discussion

### 4.1. Preclinical Insights Supporting Clinical Translation

These findings are consistent with the known role of SCFAs as mediators of gut and metabolic health, particularly propionate and butyrate, which influence lipid metabolism, inflammation, and gut barrier integrity [58,59]. Notably, the absence of TMAO elevation in SLE-F treatments contrasts with the fiber control and supports the extract’s cardiometabolic safety profile [60]. Furthermore, preserved levels of indole and carnitine—metabolites with anti-inflammatory and vascular protective roles [61]—add mechanistic plausibility to the extract’s clinical relevance.

Overall, this study demonstrates that SLE-F supplementation, particularly at the high dose (HD), modulates the gut microbiome in ways that may support metabolic and immune health. Key findings include a dose-dependent enrichment of SCFA-producing taxa (e.g., *Bifidobacterium*, *Dorea*), reductions in potentially pathogenic taxa (e.g., *Enterobacteriaceae*, *Pseudomonadota*), and enhancement of microbial pathways related to butyrate and carbohydrate metabolism. These taxonomic shifts correlated with modest changes in host markers—such as monocyte normalization and increased uric acid in HD—and functional indicators, including altered blood glucose and transit time. Together, these results support the hypothesis that microbial remodeling by SLE-F can influence metabolic and immunologic tone through interconnected mechanisms.

#### 4.1.1. Metabolic Indicators and Microbial Functional Correlates

Functional gastrointestinal disorders, such as constipation, are common worldwide, particularly among women and older adults, with prevalence estimates ranging from 2 to 28% depending on diagnostic criteria [62,63]. Although some prebiotic and probiotic interventions have shown benefits in reducing colonic transit time (CTT) and improving stool characteristics, outcomes remain inconsistent across studies [64,65,66,67]. In our study, SLE-F supplementation did not produce a uniform reduction in transit time; however, both treatment groups trended toward improved motility relative to placebo. These results suggest that microbiota-mediated effects on transit may depend on individual factors such as baseline microbial composition, dietary fiber intake, and gut physiology.

Interestingly, both SLE-F groups also exhibited a modest but statistically significant increase in blood glucose levels, while the placebo group showed a decrease. All values remained within the normal clinical range. Functional metagenomic predictions indicated upregulation of carbohydrate metabolism pathways, consistent with fucoidan’s known modulation of intestinal carbohydrate availability via α-glucosidase inhibition [68]. Given the centrality of carbohydrate intake in the Cuban diet [69,70,71,72], these findings may reflect enhanced microbial carbohydrate metabolism and host energy utilization. Conversely, the glucose-lowering effect seen in the placebo group may be partially attributable to the metabolic activity of microcrystalline cellulose (MCC), a control substance with known effects on lipid and carbohydrate regulation [73].

Uric acid levels increased across all groups, with the highest elevation in the HD group. Uric acid is the final product of purine metabolism, influenced by both dietary intake and microbial turnover [74,75,76]. During summer months, a higher consumption of purine-rich legumes and vegetables may have contributed to this trend [69,71,75]. Notably, HD-associated increases coincided with enrichment of anaerobic microbes known to participate in purine metabolism, including members of Firmicutes, Fusobacteriota, and Pseudomonadota [77,78]. These results suggest that SLE-F may influence purine metabolism indirectly through its effects on microbial community structure.

Collectively, these modest shifts in transit time, glucose, and uric acid—though not primary outcomes—highlight functional links between microbial remodeling and host metabolism. They also underscore the importance of context, including diet and baseline microbiota, in shaping individual responses to prebiotic interventions.

#### 4.1.2. Monocytes

Monocytes play a central role in innate immunity and are increasingly recognized as responsive to microbial and metabolic cues. In this study, we observed divergent trends in monocyte counts: an increase in the LD group and a decrease in HD and placebo groups. Although absolute changes were modest and remained within normal clinical limits, these shifts may reflect transient immune modulation in response to microbiome remodeling. Enrichment of taxa such as *Faecalibacterium* and *Bifidobacterium* in the HD group, along with predicted reductions in lipopolysaccharide biosynthesis pathways, may contribute to a more regulated immune tone [79,80,81]. Conversely, the rise in monocytes in the LD group could reflect a more variable response, potentially influenced by individual baseline differences or transient low-grade immune activation. While these findings are preliminary, they support further investigation into how specific microbial changes modulate innate immune parameters, such as monocyte activity.

#### 4.1.3. Quality of Life and Adverse Events

Self-reported quality of life, assessed via the SF-36 questionnaire, remained largely stable across the 90-day intervention. Among the eight domains, General Health and Vitality consistently received the lowest scores in all groups, suggesting perceived limitations in energy and overall health status. These scores likely reflect baseline characteristics of the study population or broader contextual factors rather than direct effects of the intervention.

Adverse events were predominantly mild and nonspecific, with the most frequently reported symptom being transient gastrointestinal gas—reported by over 40% of participants in both SLE-F groups versus 16% in placebo. This pattern is consistent with known effects of prebiotic fermentation [82,83]. No moderate or severe adverse events were attributed to the intervention. Other symptoms, such as joint pain, nasal congestion, or malaise, were primarily linked to intercurrent infections and seasonal viral illness. A single case of genital herpes reactivation was associated with self-reported psychological stress, rather than study product. These findings support the favorable safety and tolerability profile of SLE-F, even at higher doses, and suggest high suitability for future clinical applications.

### 4.2. Alpha Diversity

Alpha diversity metrics validated that the sequencing depth did not significantly influence richness or diversity indices, ensuring unbiased comparisons across samples. The mean baseline taxa count of 2174 ± 122 reflects substantial microbial richness, highlighting the complexity of the microbiome under study. These results confirm the reliability of the sequencing process, providing a strong foundation for downstream analyses and interpretations.

#### 4.2.1. Trends in Microbial Diversity and Potential Mechanisms of Diversity Shifts

The findings reveal clear differences in microbial diversity metrics over time among the HD, LD, and placebo groups, underscoring the limited microbiome changes observed in the absence of an active intervention [84].

The significant reductions in the Shannon diversity, particularly in the HD group, suggest selective pressures imposed by the SLE-F supplementation [82]. These changes may reflect the enrichment of specific taxa capable of metabolizing components of the intervention, leading to the competitive exclusion of others, or the suppression of certain microbes due to the intervention’s antimicrobial or prebiotic properties [85,86]. In contrast, the less pronounced changes observed in the LD group likely reflect weaker selective forces, while the overall stability in the placebo group confirms that the observed shifts were intervention related. Meanwhile, the relative stability of Simpson diversity across all groups suggests that microbial evenness remained largely intact. This indicates that the reductions in the Shannon diversity were primarily due to decreased richness—i.e., a loss in the number of taxa—rather than major changes in the relative abundance of dominant species.

#### 4.2.2. Microbiome Recovery and Resilience

The convergence of diversity metrics across groups by the EOS (Table 7, Figure 3) suggests the resilience of the gut microbiome [87]. This natural recovery may have mitigated the long-term effects of the intervention, as microbial communities stabilized over time. Additionally, adaptation to the intervention could have contributed to this stabilization, with the microbiome adjusting to the external pressures imposed by the study conditions [88].

#### 4.2.3. Clinical and Mechanistic Implications

The intervention produced significant, dose-dependent reductions in the Shannon diversity, most pronounced in the HD group, suggesting strong selective pressures on microbial richness. While reduced diversity is often linked to dysbiosis, the concurrent stability of the Simpson diversity indicates preserved evenness, potentially mitigating adverse effects. These shifts may reflect the selective enrichment of beneficial taxa capable of metabolizing the intervention. Taxonomic and functional profiling will be essential to determine whether these microbiome changes translate into improvements in immune or metabolic health. By end-of-study, diversity metrics converged across all groups, indicating microbial resilience. These findings align with previous research on prebiotics and functional polysaccharides [89,90] and highlight the importance of dose optimization to balance efficacy with preservation of microbial diversity [86,91,92].

### 4.3. Taxonomic Trends in Gut Microbiota over 90 Days of HD Treatment

The significant changes in the gut microbiota composition observed during the HD intervention over the 90-day study demonstrate the intervention’s impact on the gut microbiome.

A key finding was the significant enrichment of Actinomycetota, including the *Bifidobacteriaceae* family. Within this group, specific taxa, such as *Bifidobacterium adolescentis* and *B. longum,* exhibited marked and significant increases. These changes reflect the intervention’s ability to promote beneficial microbes associated with gut health and resilience. The enrichment of *Bifidobacterium* spp. suggests an enhancement of microbial populations known to contribute to improved gut barrier integrity and anti-inflammatory effects [93,94].

Conversely, facultative anaerobes associated with dysbiosis, such as Pseudomonadota and *Enterobacteriaceae*, exhibited notable declines, indicating a reduction in potentially pathogenic taxa linked to inflammation and gut aerobiosis. This shift suggests a positive impact of the intervention in reducing microbial groups associated with adverse health outcomes [95,96].

#### 4.3.1. Stability and Temporal Trends

The temporal dynamics of several taxa highlighted both early and sustained responses to the intervention. For instance, *Faecalibacterium*, a recognized anti-inflammatory taxon [97], demonstrated a temporary enrichment at D28 followed by a decline by the EOS. Similarly, *Porphyromonadaceae* exhibited a consistent and significant increase from 0.60% at baseline to 0.82% at D28 and maintained stability through the EOS (*p* = 0.0135). These trends reflect the intervention’s ability to promote beneficial taxa while suggesting the potential for microbiome stabilization over time [10,98].

#### 4.3.2. Microbial Health and Functional Implications

The observed shifts in the microbial composition following HD intervention indicate a transition toward a healthier and more functionally robust gut microbiome. Enrichment of short-chain fatty acid (SCFA)-producing taxa, including *Bifidobacterium*, *Dorea*, and members of Actinomycetota, suggests an enhanced microbial metabolic activity supportive of gut homeostasis [99].

SCFAs are key mediators of intestinal health, contributing to epithelial barrier integrity, immune modulation, and anti-inflammatory effects [99].

Concurrently, reductions in pro-inflammatory and potentially pathogenic taxa—such as Pseudomonadota and *Enterobacteriaceae*—further support the intervention’s role in promoting a more favorable microbial environment [100,101]. While some taxa (e.g., *Faecalibacterium*) demonstrated modest or transient responses [97,102], others like *Bifidobacterium longum* exhibited sustained increases [93,94], indicating lasting benefits for select microbial populations.

Importantly, these microbial shifts may help explain several observed host outcomes. The enrichment of the SCFA-producing taxa aligns with monocyte normalization in the HD group, suggesting potential immune-modulatory effects. Likewise, the increase in purine-metabolizing taxa may contribute to the elevated uric acid levels observed in HD, linking microbial metabolism to a host’s biochemical responses.

These findings underscore the dual action of SLE-F in enriching beneficial microbial functions while suppressing dysbiosis-associated communities. Together, these compositional and functional shifts may contribute to improved metabolic tone and immune resilience, warranting further mechanistic and longitudinal investigation [99,103,104].

#### 4.3.3. Implications for Microbiome Function and Host Response

The significant enrichment of *Bifidobacterium* spp. and the concurrent reduction in facultative anaerobes reflect the potential of HD interventions to shape the gut microbiome toward a healthier state [7,105]. These changes may translate into clinical benefits, such as enhanced gut barrier function, reduced inflammation, and improved metabolic outcomes [103]. However, the variability in response among participants (Figure 5) highlights the importance of personalized approaches in microbiome-targeted interventions [104,106].

Future studies should investigate the functional roles of enriched taxa and their contributions to host health, as well as the long-term stability of these changes. Taxonomic and functional profiling could elucidate the mechanisms underlying the observed shifts and their relevance to clinical outcomes. Additionally, examining the interplay between host factors and microbial recovery will provide insights into optimizing intervention strategies for sustained benefits [107].

#### 4.3.4. False Discovery Rate (FDR) Analysis

The FDR analysis [108] was used to evaluate changes in microbial taxa abundance over the course of the study, adjusting for multiple comparisons to reduce the likelihood of false positives. This analysis revealed both significant and non-significant shifts in response to the HD intervention. Taxa with FDR-adjusted *p*-values below 0.05—such as *Actinomycetota* and *Bifidobacterium*—showed statistically significant enrichment, supporting the idea that the intervention selectively promoted beneficial microbial populations associated with gut and metabolic health. Conversely, other taxa, such as *Pseudomonadota*, which have been associated with pro-inflammatory activity, exhibited a downward trend but did not reach statistical significance after the FDR correction. This suggests that while their decline may still contribute to the overall microbial modulation, the variability among individuals limited the ability to detect a statistically robust effect.

#### 4.3.5. Functional Predictions and Key Taxonomic Trends

The intervention induced distinct shifts in microbial taxa with important functional implications. Inflammatory taxa, including Pseudomonadota and *Enterobacteriaceae*, exhibited notable declines. For example, Pseudomonadota decreased from 4.72% at baseline to 2.7% at EOS, while *Enterobacteriaceae* dropped from 1.04% to 0.61% during the same period. These reductions align with a potential decrease in inflammation-related pathways, particularly lipopolysaccharide (LPS) production, which is commonly associated with these taxa [100,101].

In contrast, taxa involved in short-chain fatty acid (SCFA) production showed significant enrichment. Actinomycetota increased markedly from 3.04% at baseline to 9.19% at EOS (*p* = 0.00029), and *Bifidobacterium* rose from 2.10% to 7.61% (*p* = 0.00048). These increases highlight the intervention’s role in promoting taxa associated with gut health, as SCFAs such as butyrate are critical for maintaining the gut barrier integrity, regulating inflammation, and supporting host metabolism [109,110].

Other SCFA-producing taxa, such as *Dorea*, demonstrated moderate but significant increases, rising from 1.44% at the baseline to 2.08% at the EOS (*p* = 0.0053). *Faecalibacterium*, a well-known anti-inflammatory taxon and key butyrate producer, increased modestly from 0.05% at baseline to 0.12% by Day 28 and stabilized through EOS. Although the change in *Faecalibacterium* was not statistically significant (*p* = 0.1195), its upward trend suggests a beneficial response to the intervention.

While these compositional shifts provide insight into the microbiome’s potential functional capacity, it is important to note that pathway-level predictions derived from 16S rRNA gene sequencing lack the resolution to confirm strain-specific functions or metabolite production. Functional inferences based on 16S data require confirmation through metagenomic or metabolomic analyses in future studies.

#### 4.3.6. Temporal Patterns and Stability

The relative abundance trends of Actinomycetota, *Bifidobacterium, Faecalibacterium,* and *Dorea* across the baseline, Day 28, and EOS (Figure 7) underscore the intervention’s sustained impact on key taxa. The most dramatic increases were observed for Actinomycetota and *Bifidobacterium*, with significant enrichment from the baseline to EOS. These taxa showed a steady growth over time, with the largest gains occurring between Day 28 and the EOS, suggesting cumulative effects of the intervention.

*Faecalibacterium* exhibited early enrichment by Day 28, followed by stabilization, indicating a potentially rapid response to the intervention. *Dorea* displayed a significant increase between Day 28 and the EOS, further highlighting the intervention’s ability to promote SCFA-producing taxa over the long term [97].

#### 4.3.7. Translational Relevance and Research Outlook

These findings demonstrate the intervention’s ability to selectively enrich beneficial microbial taxa while reducing potentially harmful ones, suggesting a shift toward a healthier and more functional microbiome. The significant increases in SCFA-producing taxa may have clinical implications for gut health, inflammation regulation, and metabolic support. However, the variability in individual responses (Figure 7) and the non-significant changes in some key taxa, such as *Faecalibacterium*, highlight the need for further research into personalized microbiome-targeted interventions.

Future studies should explore the functional roles of enriched taxa in greater detail, using metabolomic and transcriptomic approaches to link microbial shifts to specific health outcomes. Long-term studies are also needed to assess the durability of these changes and their implications for overall health and disease prevention.

### 4.4. Microbial Ecology and Functional Inference During LD Intervention

The data reveal distinct microbial responses to the LD intervention, with significant increases in some beneficial taxa and declines in others, reflecting the complex dynamics of microbiome modulation at a lower dose.

Over the course of the 90-day LD intervention, microbial patterns suggest a subtle but biologically meaningful reshaping of the gut ecosystem. Rather than producing large-scale shifts, the intervention appears to have selectively favored the growth of certain beneficial taxa, particularly those involved in short-chain fatty acid (SCFA) production and metabolic modulation.

The emergence of *Bifidobacterium* and *Roseburia*—both well-established SCFA producers—suggests an environment increasingly supportive of anti-inflammatory and barrier-strengthening microbial functions [111,112]. These genera are commonly associated with metabolic health, gut integrity, and immune regulation, indicating that even a modest intervention may create conditions conducive to long-term resilience and host benefit [111,112].

An increase in *Collinsella* adds a unique dimension, potentially pointing to enhanced carbohydrate metabolism and bioactivation of dietary compounds, such as isoflavones. Given its known role in converting daidzein to equol, *Collinsella*’s rise may reflect improved microbial metabolic capacity with relevance to cardiovascular and hormonal health [113,114].

At the same time, a decline in *Blautia*—another carbohydrate-fermenting genus—may indicate competitive microbial dynamics, possibly reflecting shifts in substrate availability or niche occupancy [115]. Similarly, although *Faecalibacterium*, a key butyrate producer, did not significantly increase, its relative stability in the LD group may suggest that more intensive interventions are needed to support its sustained growth [116].

*Eubacterium* showed an upward trend, albeit not statistically significant, which aligns with the broader pattern of supporting SCFA-producing microbes [117]. Taken together, these trends imply that the LD intervention may promote a microbial environment geared toward functional enhancement rather than dramatic compositional restructuring. The observed shifts point to subtle but potentially beneficial changes in the microbial metabolism, with implications for both gut and systemic health.

These microbial trends in the LD group—though less dramatic than those in HD—nonetheless suggest functionally meaningful shifts. The emergence of *Bifidobacterium*, *Roseburia*, and *Collinsella* points to a microbiome increasingly oriented toward SCFA production, carbohydrate metabolism, and epithelial support. These functional improvements may underlie modest clinical trends observed in the LD group, such as monocyte elevation and stable transit time, which may reflect immune activation and gut motility effects.

Notably, the relative stability of *Faecalibacterium* and decline in *Blautia* indicate strain- or context-specific responsiveness to low-dose intervention, potentially reflecting competitive exclusion or nutrient limitation. These nuanced microbial dynamics highlight the importance of the dose in shaping both taxonomic and functional microbiome responses.

Taken together, results from both the HD and LD intervention arms reveal dose-dependent remodeling of the gut microbiome, with greater enrichment of SCFA-producing and immunoregulatory taxa in the HD group. These changes were paralleled by shifts in host markers—including blood glucose, uric acid, and monocyte levels—highlighting potential mechanistic links between microbial function and systemic outcomes. While individual variability remains an important consideration, these findings support the potential of SLE-F to modulate the gut health and metabolic tone through microbiome-mediated pathways, warranting future personalized and multi-omics studies.

#### Clinical Implications and Future Directions

Over the course of the 90-day LD intervention, microbial patterns suggest a subtle but biologically meaningful reshaping of the gut ecosystem. Rather than producing large-scale shifts, the intervention appears to have selectively favored the growth of certain beneficial taxa, particularly those involved in short-chain fatty acid (SCFA) production and metabolic modulation [59].

The emergence of *Bifidobacterium* and *Roseburia*—both well-established SCFA producers—suggests an environment increasingly supportive of anti-inflammatory and barrier-strengthening microbial functions [118]. These genera are commonly associated with metabolic health, gut integrity, and immune regulation, indicating that even a modest intervention may create conditions conducive to long-term resilience and host benefit [119].

An increase in *Collinsella* adds a unique dimension, potentially pointing to enhanced carbohydrate metabolism and bioactivation of dietary compounds, such as isoflavones [120,121]. Given its known role in converting daidzein to equol, *Collinsella*’s rise may reflect improved microbial metabolic capacity with relevance to cardiovascular and hormonal health [122].

At the same time, a decline in *Blautia*—another carbohydrate-fermenting genus—may indicate competitive microbial dynamics, possibly reflecting shifts in substrate availability or niche occupancy (F). Although *Faecalibacterium*, a key butyrate producer, did not significantly increase, its relative stability in the LD group may suggest that more intensive interventions are needed to support its sustained growth [123].

*Eubacterium* showed an upward trend, albeit not statistically significant, which aligns with the broader pattern of supporting SCFA-producing microbes [124]. Taken together, these trends imply that the LD intervention may promote a microbial environment geared toward functional enhancement rather than dramatic compositional restructuring. The observed shifts point to subtle but potentially beneficial changes in microbial metabolism, with implications for both gut and systemic health [125,126].

### 4.5. Comparisons Between High-Dose and Low-Dose Interventions and Impact of Microbiome Function

#### 4.5.1. Microbiome Composition Changes

The observed changes in the gut microbiome composition over the course of the study underscore the differential impact of the intervention dose on microbial dynamics. Participants receiving the HD treatment exhibited statistically significant shifts in the microbial composition as early as Day 28, which were sustained through the end of the study (EOS). This rapid and persistent modulation suggests that higher levels of the intervention can exert strong selective pressures on the gut microbiota, resulting in early restructuring of the microbial community. Such outcomes are consistent with prior studies demonstrating that concentrated prebiotic or polysaccharide interventions can rapidly promote the expansion of beneficial taxa and alter microbial metabolic profiles [119,127].

In contrast, the LD intervention did not produce statistically significant changes in microbial composition at either Day 28 or EOS when compared to baseline. This indicates a more limited or gradual impact on the gut microbiota, potentially reflecting insufficient substrate availability to trigger robust microbial shifts [128]. However, comparisons between the HD in the EOS cohort and the LD in the Day-28 cohort revealed significant differences, further highlighting the magnitude and earlier onset of microbiome remodeling in the HD cohort.

Interestingly, by the EOS, no significant differences were observed between the HD and LD groups. This convergence may suggest a delayed or partial response in the LD group, potentially driven by microbial adaptation over time. While these findings do not imply equivalence of outcomes, they raise the possibility that prolonged exposure—even at lower doses—can induce moderate compositional changes, albeit at a slower rate and to a lesser extent than higher doses.

Taken together, these results emphasize the dose-dependent nature of gut microbiome modulation and support previous findings that intervention strength influences both the speed and magnitude of microbial responses [129,130,131,132]. The stability of the microbiome composition in the HD group after the initial shift also points to potential resilience and durable restructuring, which may have functional implications for host metabolic or immune health.

#### 4.5.2. Functional Pathway Analysis

Functional pathway analysis revealed significant enrichment in gut health-related pathways, particularly in the HD cohorts. Key metabolic pathways—including butyrate production, short-chain fatty acid (SCFA) synthesis, and carbohydrate metabolism—were significantly elevated in the HD end-of-study (EOS) group compared to both baseline and LD cohorts. These pathways are essential for maintaining gut barrier integrity, modulating local and systemic inflammation, and supporting host energy metabolism [58,59]. Butyrate, in particular, serves as the primary energy source for colonocytes and plays a crucial role in promoting anti-inflammatory responses and epithelial health [133,134]. The observed enrichment of these microbial functions underscores the HD intervention’s capacity to enhance beneficial microbial activity and functional output.

The LD cohorts exhibited modest but consistent increases in these same pathways, indicating a dose-responsive effect on microbial metabolism. While less pronounced, these functional shifts suggest that even lower levels of the intervention can contribute positively to the gut microbial ecology over time. Notably, pathways associated with the reduction in inflammatory markers were also significantly enriched in the HD groups, pointing toward a potential systemic anti-inflammatory effect mediated through microbiome–host interactions [99,135,136].

#### 4.5.3. L/E and F/B Ratios (Figure 10)

The *Lachnospiraceae*-to-*Enterobacteriaceae* (L/E) ratio, a biomarker of improved gut inflammation, demonstrated significant increases in both LD and HD cohorts [57]. In the LD cohort, the L/E ratio increased gradually, with a significant difference observed between baseline and EOS, suggesting a steady shift favoring anti-inflammatory taxa. In the HD cohort, the L/E ratio rose rapidly by Day 28 and stabilized through EOS, indicating an earlier and more robust response.

The Firmicutes-to-Bacteroidetes (F/B) ratio increased, but not significantly in both cohorts, reflecting a favorable shift in microbial balance [137]. These changes highlight the intervention’s potential to modulate microbial composition toward a healthier state, with HD treatments inducing faster and more pronounced effects.

#### 4.5.4. Dose-Dependent Effects and Optimization Strategies

The findings emphasize the dose-dependent effects of the intervention on gut microbiome composition and functionality [84]. The HD treatment consistently demonstrated greater enrichment of beneficial taxa, functional pathways, and inflammation-related biomarkers. These results suggest that HD interventions may be more effective for achieving early and sustained improvements in gut health. However, the gradual improvements observed in the LD cohort indicate that lower doses may still offer long-term benefits, albeit at a slower pace.

Future studies should explore extending treatment durations for the LD group to determine whether it can achieve similar outcomes to the HD intervention over time. Additionally, combining lower doses with other prebiotics or dietary modifications could amplify their impact. Personalized approaches based on baseline microbiome profiles may also optimize dosing strategies, ensuring that individuals receive the most effective intervention tailored to their microbiome composition and health needs. Collectively, these results emphasize the critical role of dose optimization in achieving a balance between therapeutic efficacy, safety, and accessibility in microbiome-based interventions.

### 4.6. Limitations of the Study

This study provides meaningful insights into the dose-dependent impact of the intervention on gut microbiota composition and functionality; however, several limitations must be acknowledged to properly contextualize the findings. These include the relatively short intervention period, modest sample size, and reliance on predictive functional analyses rather than direct metabolic measurements. Moreover, the lack of dietary control and the focus on a healthy adult population may limit the generalizability of the results.

#### 4.6.1. Sample Size and Cohort Variability

The relatively small sample size within the HD and LD cohorts may limit the generalizability of the results. Inter-individual variability in the microbiome composition and response to the intervention further complicates the interpretation of findings, as outliers could disproportionately influence observed trends. Larger, more diverse cohorts are needed to confirm these results and improve statistical power.

#### 4.6.2. Short Study Duration

The study duration of 90 days may not have been sufficient to capture the full extent of microbiome recovery or stabilization, particularly for slower-responding taxa such as *Faecalibacterium*. Extending the duration of the intervention and follow-up periods would provide better insights into the long-term effects and sustainability of the observed changes.

#### 4.6.3. Limited Functional Analysis

While the functional pathway analysis identified significant enrichment in pathways associated with gut health, such as SCFA production and carbohydrate metabolism, these predictions were inferred from taxonomic data rather than direct metagenomic or metabolomic measurements. Future studies should incorporate multi-omics approaches to validate functional findings and elucidate the mechanistic underpinnings of microbial shifts.

#### 4.6.4. Lack of Dietary Control

Although the intervention significantly influenced the microbiome, dietary variations among participants were not tightly controlled. As the diet is a major determinant of gut microbiome composition, differences in dietary intake may have confounded the results. Future studies should incorporate standardized dietary protocols or detailed dietary tracking to isolate the intervention’s effects.

#### 4.6.5. Potential for Unmeasured Confounders

Other factors, such as medication use, baseline health status, or lifestyle differences, were not fully controlled or accounted for in the study. These variables may have influenced individual responses to the intervention, and their impact should be evaluated in future research.

#### 4.6.6. Focus on Dose Without Other Combinations

The study primarily evaluated LD and HD interventions without exploring potential synergistic effects of combining the intervention with prebiotics, dietary modifications, or multi-strain probiotics. These combinations could amplify the observed effects and warrant investigation in future studies.

#### 4.6.7. Limited Taxonomic Resolution

While the study analyzed changes at the genus and family levels, species-level resolution was not consistently achieved. This limits the ability to identify specific microbial species responsible for functional changes. Advanced sequencing technologies and strain-specific analyses would enhance the granularity of future findings.

Addressing these limitations in future research will strengthen the evidence base for the intervention’s effects on the gut microbiome and its clinical relevance. Larger, longer-term studies that incorporate multi-omics approaches, standardized dietary protocols, and direct measurements of host health outcomes are critical for optimizing microbiome-targeted interventions and translating these findings into practical applications.

## 5. Conclusions

The clinical trial results for SLE-F demonstrate that it is both safe and effective in modulating the gut microbiome. These findings underscore a favorable safety profile while positioning SLE-F as a promising prebiotic candidate for addressing dysbiosis and promoting pathways critical for gut and systemic health. Notably, the intervention enriched short-chain fatty acid (SCFA)-producing taxa (e.g., *Bifidobacterium*, *Dorea*), reduced potentially pro-inflammatory taxa (e.g., *Enterobacteriaceae*, *Pseudomonadota*), and was associated with modest host shifts, including glucose and uric acid modulation.

Importantly, functional inferences based on 16S-predicted pathways require confirmation through metagenomic or metabolomic studies to validate the microbial activity at the species and metabolite level. Future research should prioritize these deeper functional analyses, long-term studies, and personalized intervention strategies to better link microbiome remodeling with durable host outcomes. This study provides a robust foundation for the continued development of safe, targeted prebiotic interventions to restore and sustain microbial homeostasis.

## Figures and Tables

**Figure 1 microorganisms-13-01545-f001:**
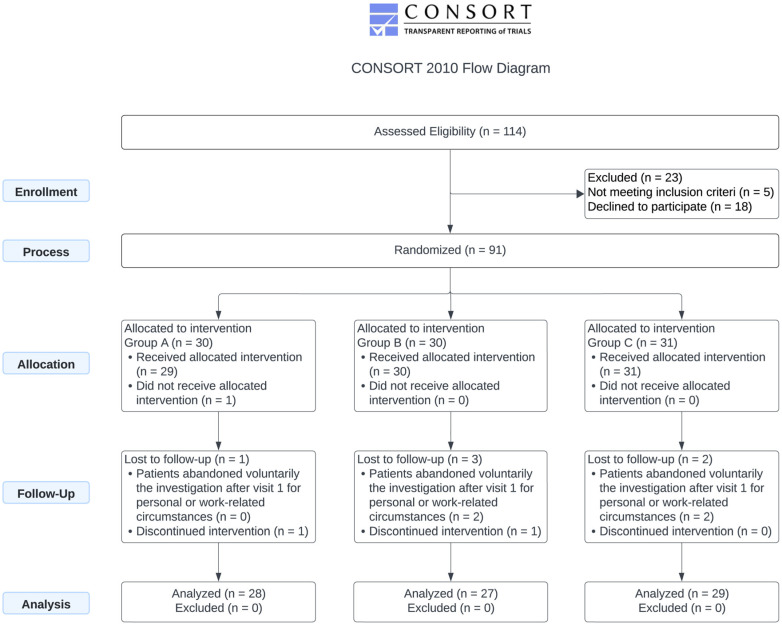
Consort flow diagram. The study involved the administration of 125 mg (LD), 500 mg of SLE-F (HD), or a 125 mg placebo capsule (P) every 12 h for 90 days.

**Figure 2 microorganisms-13-01545-f002:**
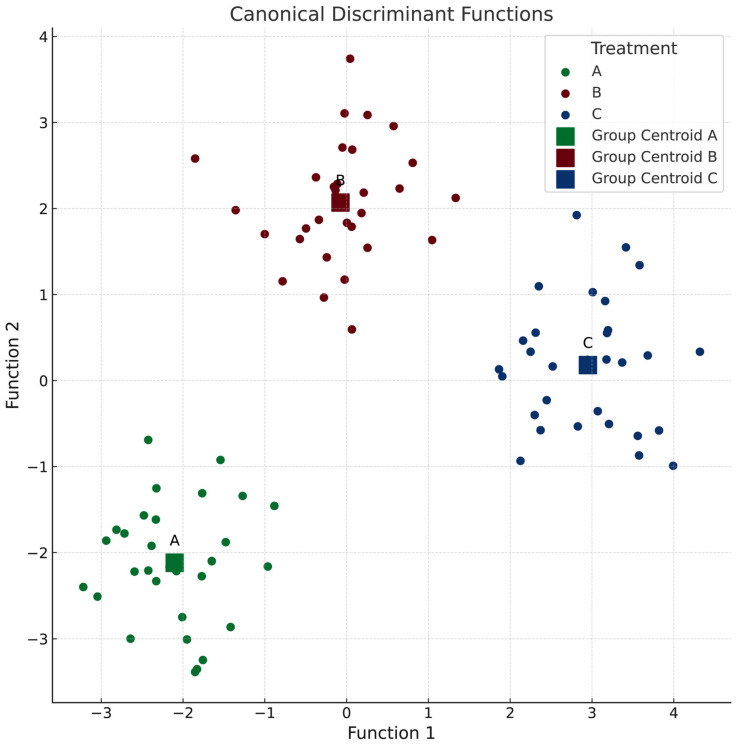
Discriminant analysis to identify patterns between groups. Function 1 includes the following: glucose, MCHC, MCH, HbA1c, calprotectin, neutrophils, MCV, HTC, RDW-SD, hemoglobin, eosinophils, triglycerides, and RBC. Function 2 includes the following: uric acid, intestinal transit time, monocytes, ALAT, WBC, cholesterol, ASAT, creatinine, lymphocytes, C-reactive protein, platelets, MPV, urea, and RDW-CV.

**Figure 3 microorganisms-13-01545-f003:**
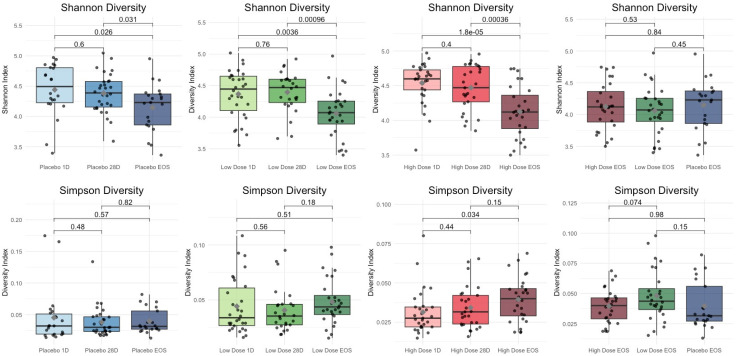
Shannon and Simpson diversity across cohorts.

**Figure 4 microorganisms-13-01545-f004:**
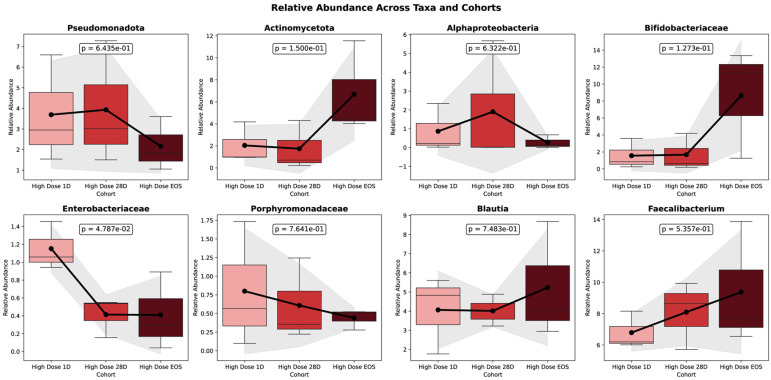
Relative abundance of key taxa across time points in HD cohort.

**Figure 5 microorganisms-13-01545-f005:**
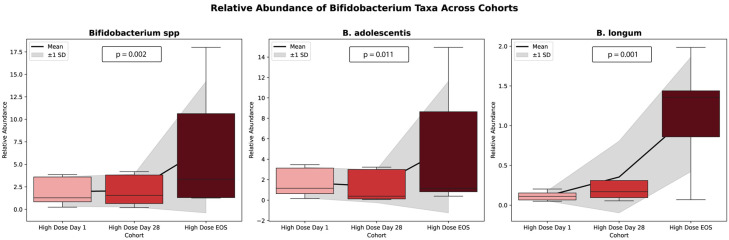
Trends in *Bifidobacterium* spp. relative abundance across study phases for the high dose cohort. This chart illustrates the relative abundance of *Bifidobacterium* spp. across three time points: baseline (1), Day 28 (28), and end of study (EOS). Each line represents a unique participant, with the average trend across all participants denoted by the bolded mean line. The graph highlights the variability in *Bifidobacterium* abundance responses among participants and overall increases in abundance over time.

**Figure 6 microorganisms-13-01545-f006:**
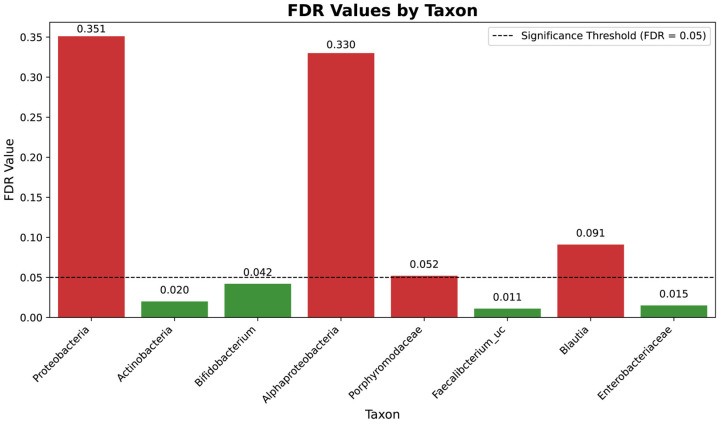
False Discovery Rate (FDR) analysis of taxonomic shifts across study phases for the high-dose cohort. The figure displays the FDR-adjusted *p*-values for the taxa analyzed, summarizing the significance of changes in microbial abundance across the baseline, Day 28, and the end of study (EOS). False Discovery Rate (FDR) analysis of taxonomic shifts across study phases for the high-dose cohort. The figure displays the FDR-adjusted *p*-values for the taxa analyzed, summarizing the significance of changes in microbial abundance across the baseline, Day 28, and the end of study (EOS). Taxa with FDR values below 0.05 (green bars) are highlighted as significantly enriched or depleted, while those with higher FDR values (red bars), such as Pseudomonadota, are considered non-significant. This visualization emphasizes the intervention’s differential impact on microbial taxa.

**Figure 7 microorganisms-13-01545-f007:**
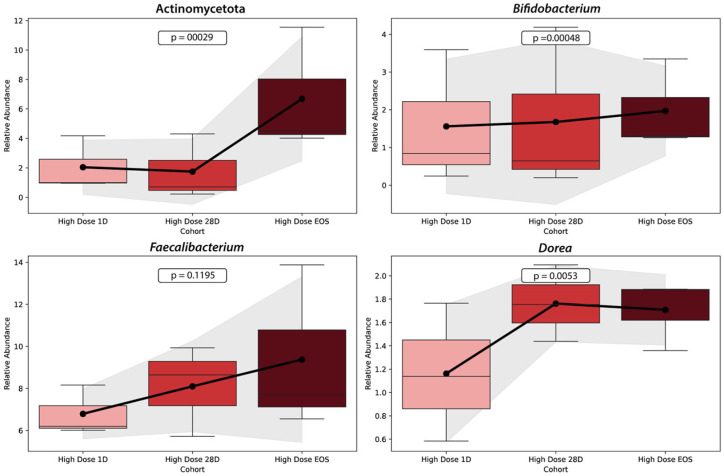
Changes in the relative abundance of key microbial taxa across study phases in the HD Cohort. This figure shows the relative abundance changes in four microbial taxa (*Actinomycetota*, *Bifidobacterium*, *Faecalibacterium*, and *Dorea*) over three time points: baseline (Day 1), Day 28, and end of study (EOS) in the HD cohort. Each panel presents the trends for a specific taxon, with boxplots displaying the data distribution and lines connecting mean values to illustrate the overall trajectory. The graph was generated using data on microbial relative abundances from the HD cohort at three time points. Statistical analyses, including *p*-values for trends over time, were calculated to identify significant changes in each taxon’s abundance. The visualization was performed using Python’s Seaborn v0.13.2 and Matplotlib v3.10.3 libraries, with boxplots representing variability within the cohort and shaded regions around the mean lines indicating ±1 standard deviation.

**Figure 8 microorganisms-13-01545-f008:**
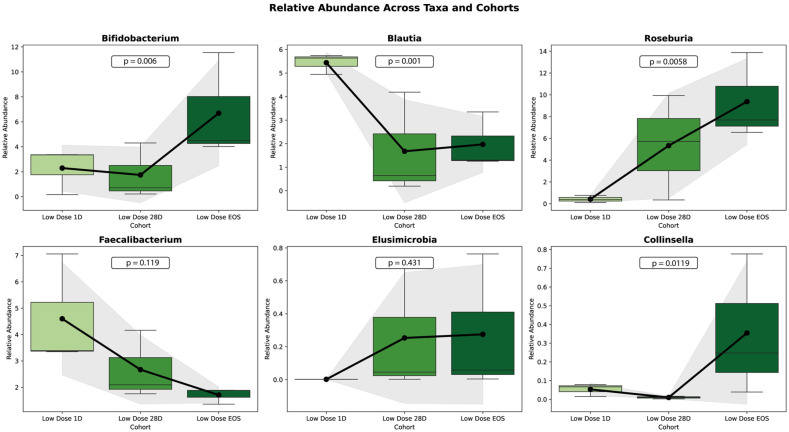
Trends in the relative abundance of key microbial taxa in the LD cohort across study phases. The relative abundances of six key taxa across three cohorts (low-dose baseline ID, low-dose 28D, and low-dose EOS) are shown as boxplots. Each boxplot displays the median (horizontal line), interquartile range (box), and outliers (dots). Overlaid are trendlines representing the mean values for each cohort (black line) and shaded areas indicating ±1 standard deviation (gray). Annotated *p*-values represent the statistical significance of pairwise comparisons between cohorts, calculated using a *t*-test.

**Figure 9 microorganisms-13-01545-f009:**
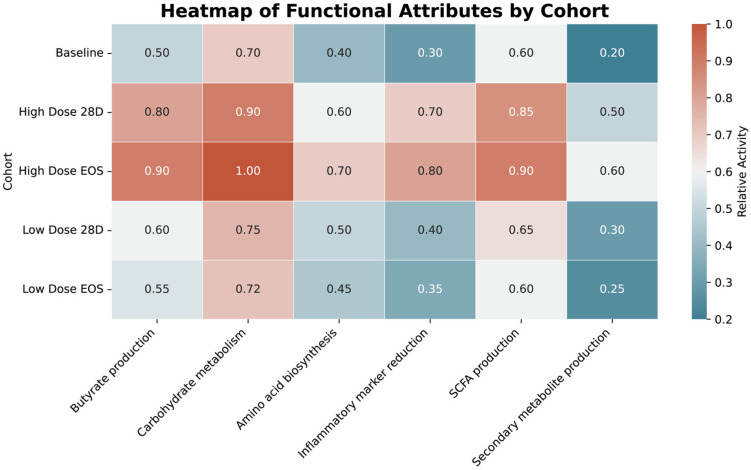
Heatmap of functional pathway abundance across cohorts. The heatmap visualizes the relative abundance of functional pathways across cohorts, with warmer colors (red/orange) indicating higher abundance and cooler colors (blue) indicating lower abundance. The numerical values represent pathway activity levels, emphasizing significant enrichment in HD cohorts, particularly in pathways related to SCFA production, butyrate production, and carbohydrate metabolism.

**Figure 10 microorganisms-13-01545-f010:**
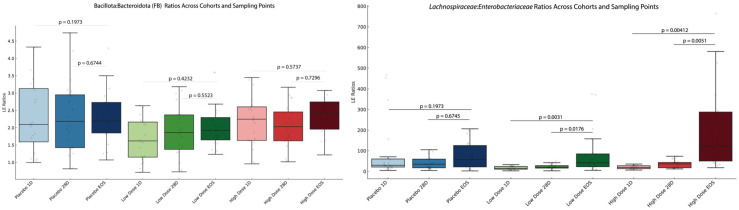
L/E and F/B ratios for low- and high-dose cohorts across study phases. The figure illustrates the changes in L/E and F/B ratios across three study phases—baseline, Day 28 (28D), and end of study (EOS)—for P, LD, and HD cohorts. The left panel represents the F/B ratios for each cohort at three different sampling times. The right panel represents these results obtained for the L/E ratios. Statistical significance is indicated by the annotated *p*-values, highlighting changes over time within each cohort.

**Table 1 microorganisms-13-01545-t001:** Demographic characteristics of the study population.

		Treatment		
		LD	HD	*p*
		N * = 29	N = 30	N = 31
Sex	Female	13 (44.8%)	19 (63.3%)	18 (58.4%)
	Male	16 (55.2%)	11 (36.7%)	13 (41.9%)
Age (years)	Average (SD)	37.6 (14.2)	43.0 (16.6)	42.1 (13.7)
	Median value (RI)	34.0 (28.0)	49.0 (31.0)	50.0 (27.0)
	(Minimum; Maximum)	(21; 69)	(19; 67)	(21; 60)
Weight (Kg)	Average (SD)	65.6 (11.6)	66.4 (13.7)	68.9 (15.4)
	Median (RI)	67.0 (13.4)	65.0 (14.5)	66.5 (11.1)
	(Minimum; Maximum)	(39.7; 92.8)	(42.6; 101.5)	(42.2; 122.1)
Height (cm)	Average (SD)	169.3 (8.2)	166.0 (11.2)	170.7 (8.1)
	Median (RI)	169.0 (11.8)	161.5 (18.9)	170.5 (14.0)
	(Minimum; Maximum)	(149.0; 186.5)	(145.0; 187.5)	(157.0; 188.0)
BMI (Kg/m^2^)	Average (SD)	22.9 (3.7)	24.0 (3.8)	23.6 (4.8)
	Median (RI)	22.6 (4.9)	23.8 (5.4)	22.4 (3.7)
	(Minimum; Maximum)	(15.8; 30.2)	(16.6; 32.0)	(17.1; 38.3)

* N indicates the number of individuals in the cohort.

**Table 2 microorganisms-13-01545-t002:** Treatment interruptions.

		Treatment		
		LD	HD	P
		N = 29	N = 30	N = 31
Interruption	Yes	1 (3.4%)	3 (10.0%)	2 (6.5%)
	No	28 (96.6%)	27 (90.0%)	29 (93.5%)
Cause of interruption	Adverse Event	1 (3.4%) AE	1 (3.3%) AE	0 (0.0%)
	Voluntary abandonment	0 (0.0)	2 (6.7%)	2 (6.5%)

**Table 3 microorganisms-13-01545-t003:** Viral infection during study.

		Treatment			*p*-Value
		LD	HD	P	(χ^2^)
		N = 29	N =27	N = 28	
Viral infection during the study	No	15 (51.7%)	14 (46.7%)	14 (45.2%)	0.869
	yes	14 (48.3%)	16 (53.3%)	17 (52.2%)	
Period of infection	Before/Beginning	4 (28.6%)	6 (37.5%)	10 (58.8%)	0.207
	During	10 (71.4%)	10 (62.5%)	7 (41.2%)	

**Table 4 microorganisms-13-01545-t004:** Classification function coefficients.

Metric.		Treatment						*p*-Value (ANOVA)
	Normal Reference Range	LD		HD		P		
		Mean	SD	Mean	SD	Mean	SD	
Calprotectin (0–90)	<160 μg/g	13.9	120.2	79.6	171.1	79.6	204.2	0.480
Neutrophils (0–90)	1.42–6.34 × 10^9^/L	−0.1	1.1	0.2	1.4	0.4	1.5	0.620
Hemoglobin (0–90)	F:123–153/M:140–175 g/L	−0.7	4.1	−1.3	5.2	−2.5	9.0	0.761
RBC (0–90)	F:4.1–5.1/M:4.5–5.9 cels/μL	−0.1	0.2	0.0	0.2	0.0	0.2	0.965
HTC (0–90)	F: 0.35–0.47/M:0.40–0.52 L/L	0.0	0.0	0.0	0.0	0.0	0.0	0.531
MCV (0–90)	80–96 fl	1.2	7.2	1.1	2.5	3.2	4.9	0.560
MCH (0–90)	28–33 pg/cell	0.0	0.3	0.0	0.5	−0.4	1.2	0.297
MCHC (0–90)	33–36 g/dL	−0.4	2.2	−0.4	0.6	−1.5	1.3	0.137
RDW-SD (0–90)		1.0	6.1	0.7	2.2	2.5	5.2	0.623
RDW-CV (0–90)		0.2	0.8	0.0	0.5	0.2	0.9	0.669
Platelets (0–90)	150–450 10^3^/uL	0.7	47.1	−5.8	45.1	21.4	45.1	0.307
MPV (0–90)	F: 12–16/M: 14–17.4 g/dL	0.5	1.0	0.3	0.7	0.7	0.9	0.518
Intestinal transit time (0–90)		3.0	9.7	11.1	19.1	−10.2	27.1	0.027
CRP (0–90)	<5 mg/L	−0.3	0.8	0.1	1.1	−0.4	1.3	0.524
Glucose (0–90)	3.3–6.1 mmol/L	−0.4	0.4	−0.1	0.7	0.6	0.5	0.000
Creatinine (0–90)	47.6–113.4 µmol/L	−1.6	7.4	−3.4	6.9	1.6	6.6	0.209
Urea (0–90)	<8.3 mmol/L	0.2	0.9	−0.1	1.3	0.1	0.9	0.641
ALAT (0–90)	<45 U/L	0.9	6.6	−3.5	6.3	2.6	11.4	0.153
ASAT (0–90)	<40 U/L	1.3	8.8	−3.8	7.8	−2.1	5.3	0.203
Cholesterol (0–90)	<5.2 mmol/L	0.1	0.6	−0.3	0.6	−0.1	0.4	0.186
Triglycerides (0–90)	0.46–1.8 mmol/L	0.1	0.4	0.1	0.4	0.1	0.2	0.916
Uric acid (0–90)	119–464 mmol/L	−20.4	27.2	−60.4	37.4	−26.4	30.4	0.004
WBC (0–90)	4.4–11.3 × 10^9^/L	0.5	1.0	−0.1	1.4	1.1	1.8	0.118
Lymphocytes (0–90)	0.71–4.53 × 10^9^/L	0.6	0.6	0.3	0.3	0.4	0.7	0.336
Monocytes (0–90)	2–8%	−1.5	3.9	3.4	4.9	1.5	6.6	0.046
Eosinophils (0–90)	2–4%	0.1	1.9	−0.7	4.1	−0.6	4.3	0.819
Basophils (0–90)	0–1%	−1.0	3.3	−1.8	2.2	−0.7	4.6	0.707
HbA1c (0–90)	<5.7%	0.1	0.2	0.0	0.2	0.0	0.2	0.280

Abbreviations used: RBC: Red blood cell (red blood cell count); HTC: hematocrit; MCV: mean corpuscular volume; MCH: mean corpuscular hemoglobin; MCHC: mean corpuscular hemoglobin concentration; RDW-SD: red cell distribution width—standard deviation; RDW-CV: red cell distribution width—coefficient of variation; MPV: mean platelet volume; CRP: C-reactive protein; ALAT: alanine aminotransferase; ASAT: aspartate aminotransferase; WBC: white blood cell; HbA1c: glycated hemoglobin.

**Table 5 microorganisms-13-01545-t005:** Body weight and body mass index.

Body Weight (Kg)		Treatment			*p*-Value (Kruskal–Wallis)
		LD	HD	P	
		N = 29	N =27	N = 28	
Baseline	Average (SD)	65.0 (11.4)	66.9 (14.0)	69.0 (15.7)	
	Median (RI)	66.8 (13.6)	65.1 (14.6)	66.9 (12.8)	0.426
	(Minimum; Maximum)	(39.7; 92.8)	(42.6; 101.5)	(42.2; 122.1)	
Day 30	Average (SD)	65.7 (10.8)	68.2 (13.5)	69.6 (15.9)	
	Median (RI)	67.8 (12.3)	66.7 (13.5)	67.4 (13.3)	0.176
	(Minimum; Maximum)	(41.3; 90.9)	(47.6; 103.8)	(41.6; 122.5)	
Day 90	Average (SD)	65.3 (10.5)	67.9 (14.0)	69.4 (15.7)	
	Median (RI)	67.9 (13.5)	67.3 (15.2)	67.5 (12.3)	0.264
	(Minimum; Maximum)	(41; 89.1)	(47.7; 104)	(41.7; 121.8)	
P (Wilcoxon)	0–30	0.063	0.002	0.011	
	30–90	0.959	0.904	0.249	
	0–90	0.171	0.012	0.099	
**Body Mass Index (BMI)**		**Treatment**			***p*-Value (Kruskal–Wallis)**
		**LD**	**HD**	**P**	
		**N = 29**	**N =27**	**N = 28**	
Baseline	Average (SD)	22.8 (3.7)	24.1 (3.9)	23.6 (4.9)	
	Median (RI)	22.6 (4.6)	23.8 (5.2)	22.5 (3.9)	0.426
	(Minimum; Maximum)	(15.8; 30.2)	(16.6; 32)	(17.1; 38.3)	
Day 30	Average (SD)	23.0 (3.4)	24.6 (3.5)	23.8 (5.0)	
	Median (RI)	22.6 (4.0)	24.5 (4.6)	22.7 (3.9)	0.176
	(Minimum; Maximum)	(16.8; 30.3)	(18.4; 32)	(16.9; 38.4)	
Day 90	Average (SD)	22.9 (3.4)	24.5 (3.8)	23.8 (4.9)	
	Median (RI)	22.5 (4.8)	24.4 (5.6)	22.6 (2.9)	0.264
	(Minimum; Maximum)	(16.6; 30.4)	(17.4; 32.1)	(16.9; 38.2)	
P (Wilcoxon)	0–30	0.055	0.002	0.014	
	30–90	0.990	1.000	0.245	
	0–90	0.156	0.015	0.088	

**Table 6 microorganisms-13-01545-t006:** Dimensions of the SF-36 Health Questionnaire.

		Treatment						*p*-Value (ANOVA)
		LD		HD		P		
Time Period	Dimension	Mean	SD	Mean	SD	Mean	SD	
0	Physical Function	97.4	3.7	93.4	9.2	94.3	8.8	0.122
	Physical Role	92.5	9.2	83.8	17.3	91.1	16.0	0.070
	Body Pain	83.2	15.4	73.5	20.8	79.2	24.0	0.221
	General Health	77.4	14.5	81.4	16.7	75.7	18.1	0.450
	Vitality	70.5	17.7	70.0	23.5	61.6	19.1	0.194
	Social Function	91.8	12.6	90.5	15.4	88.0	17.5	0.635
	Emotional Role	90.2	16.4	88.0	22.6	91.3	17.7	0.814
	Mental Health	81.2	15.1	80.4	17.6	80.6	18.2	0.983
90	Physical Function	97.0	3.9	94.0	8.8	95.2	7.4	0.296
	Physical Role	90.6	12.9	86.7	16.7	91.4	14.4	0.476
	Body Pain	82.6	15.3	78.0	19.9	82.4	21.4	0.619
	General Health	78.9	11.5	81.0	16.5	76.1	16.5	0.499
	Vitality	71.0	18.3	72.2	21.2	67.6	17.6	0.657
	Social Function	91.5	13.6	90.5	16.3	90.7	16.1	0.968
	Emotional Role	91.4	14.8	90.0	20.1	89.5	19.3	0.925
	Mental Health	81.1	16.3	82.2	16.8	83.3	18.2	0.887

**Table 7 microorganisms-13-01545-t007:** Adverse events observed in the study.

		Treatment			*p*-Value (χ^2^)
		LD	HD	P	
		29	30	31	
AE		31	26	8	
AE presence	Yes	18 (62.1%)	17 (56.7%)	5 (16.1%)	0.0002
	No	11 (37.9%)	13 (43.3%)	26 (83.9%)	
Intestinal pain or discomfort		5 (17.2%)	2 (6.7%)	1 (3.2%)	
Gas		12 (41.4%)	14 (46.7%)	5 (16.1%)	
Joint pain		0	1 (3.3%)	0	
Nasal obstruction		0	1 (3.3%)	0	
Diarrhea		5 (17.2%)	4 (13.3%)	0	
Constipation		5 (17.2%)	1 (3.3%)	1 (3.2%)	
Dizziness or nausea		0	1 (3.3%)	0	
General malaise		1 (3.4%)	0	1 (3.2%)	
Headache		4 (13.8%)	1 (3.3%)	1 (3.2%)	
Pain after taking capsules		1 (3.4%)	0	0	
Migraine		1 (3.4%)	0	0	
Itching/Skin irritation		2 (6.9%)	1 (3.3%)	0	
Abdominal distension		0	1 (3.3%)	0	
Genital herpes		0	1 (3.3%)	0	

**Table 8 microorganisms-13-01545-t008:** Comparison of Shannon and Simpson diversity indices across cohorts and time points using Tukey’s honest significant difference test [55].

Group 1	Group 2	Mean Difference	*p*-adj	Lower	Upper	Reject	Metric
High Dose Baseline	High-Dose 28D	−0.0689	0.7036	−0.2744	0.1365	FALSE	Shannon
High Dose Baseline	High-Dose EOS	−0.4155	0	−0.6229	−0.2082	TRUE	
High-Dose 28D	High-Dose EOS	−0.3466	0.0004	−0.5539	−0.1393	TRUE	
Placebo Baseline	Placebo 28D	−0.0573	0.8531	−0.3126	0.198	FALSE	
Placebo Baseline	Placebo EOS	−0.2925	0.0365	−0.57	−0.015	TRUE	
Placebo 28D	Placebo EOS	−0.2352	0.0824	−0.494	0.0236	FALSE	
High-Dose EOS	Low-Dose EOS	−0.0637	0.8058	−0.3066	0.1792	FALSE	
High-Dose EOS	Placebo EOS	0.0229	0.9762	−0.2394	0.2853	FALSE	
Low-Dose EOS	Placebo EOS	0.0866	0.7106	−0.1757	0.3489	FALSE	
Low Dose Baseline	Low-Dose 28D	0.0287	0.9533	−0.2042	0.2617	FALSE	
Low Dose Baseline	Low-Dose EOS	−0.3063	0.0055	−0.5348	−0.0779	TRUE	
Low-Dose 28D	Low-Dose EOS	−0.3351	0.0032	−0.5719	−0.0983	TRUE	
High Dose Baseline	High-Dose 28D	0.0028	0.718	−0.0058	0.0114	FALSE	Simpson
High Dose Baseline	High-Dose EOS	0.0081	0.0744	−0.0006	0.0168	FALSE	
High-Dose 28D	High-Dose EOS	0.0053	0.3227	−0.0034	0.014	FALSE	
Placebo Baseline	Placebo 28D	−0.0072	0.668	−0.0274	0.0129	FALSE	
Placebo Baseline	Placebo EOS	−0.0058	0.8007	−0.0277	0.0161	FALSE	
Placebo 28D	Placebo EOS	0.0014	0.9851	−0.019	0.0218	FALSE	
High-Dose EOS	Low-Dose EOS	0.0087	0.1841	−0.003	0.0204	FALSE	
High-Dose EOS	Placebo EOS	0.0001	0.9997	−0.0125	0.0128	FALSE	
Low-Dose EOS	Placebo EOS	−0.0086	0.242	−0.0212	0.0041	FALSE	
Low Dose Baseline	Low-Dose 28D	−0.0035	0.8237	−0.0178	0.0107	FALSE	
Low Dose Baseline	Low-Dose EOS	0.004	0.7737	−0.01	0.018	FALSE	
Low-Dose 28D	Low-Dose EOS	0.0075	0.4311	−0.0069	0.022	FALSE	

**Table 9 microorganisms-13-01545-t009:** Changes in the relative abundance of key microbial taxa in the HD cohort and their inferred functional roles across study phases.

	Mean Relative Abundance			
Taxon	Baseline	28D	EOS	Functional Role
Pseudomonadota	3.8	4.0	2.2	Inflammation (LPS)
Actinomycetota	2.1	2.0	7.1	SCFA production
*Bifidobacterium*	1.8	1.39	8.8	SCFA production
*Faecalibacterium*	6.6	8.1	9.4	SCFA production
*Enterobacteriaceae*	1.9	0.4	0.4	Inflammation (LPS)
*Porphyromonadaceae*	0.8	0.6	0.5	Carbohydrate metabolism
*Blautia*	4.1	4.0	5.2	Carbohydrate metabolism
*Dorea*	1.4	1.4	2.0	SCFA production

**Table 10 microorganisms-13-01545-t010:** Statistical comparisons of microbiome changes across treatment groups and time points.

Group 1	Group 2	*p*-Value	Significant?
Baseline	High-Dose 28D	*p* < 0.05	Yes
Baseline	High-Dose EOS	*p* < 0.05	Yes
Baseline	Low-Dose 28D	*p* > 0.05	No
Baseline	Low-Dose EOS	*p* > 0.05	No
High-Dose 28D	High-Dose EOS	*p* > 0.05	No
High-Dose 28D	Low-Dose 28D	*p* > 0.05	No
High-Dose 28D	Low-Dose EOS	*p* > 0.05	No
High-Dose EOS	Low-Dose 28D	*p* < 0.05	Yes
High-Dose EOS	Low-Dose EOS	*p* > 0.05	No
Low-Dose 28D	Low-Dose EOS	*p* > 0.05	No

## Data Availability

The original contributions presented in this study are included in the article/Appendix A. Further inquiries can be directed to the corresponding author.

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
