# Peer review of "Gut Microbiome Modulation and Health Benefits of a Novel Fucoidan Extract from Saccharina latissima: A Double-Blind, Placebo-Controlled Trial"

_microorganisms, 2025, doi:10.3390/microorganisms13071545_

Round 1

Reviewer 1 Report

Comments and Suggestions for Authors

The present study evaluated the effects of SLE-F on the gut microbiome modulation, evidencing its properties as a promising therapeutic agent for addressing dysbiosis, increasing gut microbial diversity, and promoting gut health.

The objectives of the current paper but also of the study’s objectives should be clearly highlighted both in the abstract and at the end of the introduction.

-Did the authors perform a systematic or at least an exhaustive literature search before designing their study and to write the introduction section of the manuscript?

-It is stated in the manuscript that "Participants were withdrawn from the study if they experienced intolerable adverse effects, developed comorbid conditions, or consumed probiotic or prebiotic products other than the investigational substance during the study period." Is the retained number of subjects participating in the study representative for each group?

-Is the study protocol deposited and available online for consultation? For example in OSF registries or similar?

-Is the "Informed Consent Statement" form available as supplementary material?

-Did the authors also use a microbial-culture dependent approach? It would be interesting to know which microorganisms are viable at each time point.

Author Response

REVIEWER 1

The present study evaluated the effects of SLE-F on the gut microbiome modulation, evidencing its properties as a promising therapeutic agent for addressing dysbiosis, increasing gut microbial diversity, and promoting gut health.

The objectives of the current paper but also of the study’s objectives should be clearly highlighted both in the abstract and at the end of the introduction.

-Did the authors perform a systematic or at least an exhaustive literature search before designing their study and to write the introduction section of the manuscript?

RESPONSE: We appreciate the reviewer’s observation regarding the need for clarity in the study’s objectives and literature foundation.

Objectives Clarification: We have revised both the abstract and the end of the introduction to clearly and concisely state the primary objectives of the study as well as the aims of the current analysis. These changes highlight the scientific rationale and anticipated contribution of our work.

We updated the Abstract as follows: Abstract: This randomized, double-blind, placebo-controlled, three-arm clinical trial evaluated the effects of a proprietary bioactive fucoidan-rich extract derived from Saccharina latissima (SLE-F) on gut microbial composition and function in healthy adults. The objective of the study was to assess the potential of SLE-F to beneficially modulate the gut microbiome, with this paper specifically reporting on microbial diversity, taxonomic shifts, and functional pathway outcomes. Ninety-one participants received either a low dose (125 mg), high dose (500 mg), or placebo twice daily for four weeks. The primary endpoint was microbiome composition assessed via 16S rRNA sequencing (V3–V4 region), with secondary outcomes including surveys, adverse event monitoring, and clinical evaluations. High-dose supplementation resulted in dose-dependent improvements in microbial diversity, increased abundance of beneficial taxa including Bifidobacterium, Faecalibacterium, and Lachnospiraceae, and reductions in inflammation-associated taxa such as Enterobacteriaceae and Pseudomonadota. Functional pathway analysis showed enhancement in short-chain fatty acid biosynthesis and carbohydrate metabolism. The low-dose group showed modest benefits, primarily increased Bifidobacterium, with limited functional changes. In vitro colonic simulations further demonstrated a dose-dependent increase in short-chain fatty acids and postbiotic metabolite production following SLE-F exposure. SLE-F was well tolerated, with only mild, nonspecific adverse events reported. These findings support the potential of SLE-F as a safe and effective microbiome-modulating agent, warranting further study of long-term use and synergy with dietary interventions.

RESPONSE: We have modified the introduction to include additional pertinent reference and the following paragraph at the end of the introduction section: “This report presents an in-depth analysis of the microbiome-related outcomes of a randomized, placebo-controlled clinical trial investigating SLE-F supplementation in healthy adults. The primary objective of the overall study was to assess the impact of SLE-F on gut microbial composition and functional potential over a three-month intervention. Specifically, this paper presents a detailed examination of microbiota diversity, taxonomic shifts, and predicted microbial functions to evaluate the prebiotic potential of SLE-F and its implications for metabolic and immune health. These findings build upon earlier mechanistic insights and provide a clinical basis for future therapeutic applications in at-risk or dysbiotic populations.”

RESPONSE: Literature Review Scope: We confirm that a thorough and structured literature search was performed prior to designing the study and writing the manuscript. However, in the introduction, we intentionally included only the most germane references to maintain a focused narrative and to avoid redundancy. We have now added a brief statement to the introduction to reflect this approach and, where appropriate, incorporated a few additional key citations to address this concern.

-It is stated in the manuscript that "Participants were withdrawn from the study if they experienced intolerable adverse effects, developed comorbid conditions, or consumed probiotic or prebiotic products other than the investigational substance during the study period." Is the retained number of subjects participating in the study representative for each group?

RESPONSE: We appreciate the reviewer’s question. As shown in the CONSORT flow diagram (Figure 1), five individuals were excluded prior to randomization due to not meeting the study’s inclusion criteria. These exclusions occurred during the screening phase and are transparently documented in the figure. Accordingly, the final number of participants per group reflects only those who were eligible, randomized, and assigned to the intervention arms, ensuring group representativeness and minimizing potential bias.

-Is the study protocol deposited and available online for consultation? For example in OSF registries or similar?

RESPONSE: We appreciate the reviewer’s inquiry regarding public access to the study protocol. As noted in Section 2.4 of the manuscript, the clinical trial is registered in the Cuban Public Registry of Clinical Trials (RPCEC) under the identifier RPCEC00000443, where the full study protocol is publicly available for consultation. The RPCEC is a WHO-recognized primary registry and is listed within the World Health Organization’s International Clinical Trials Registry Platform (ICTRP), ensuring global standards for trial transparency and accessibility. The registry can be accessed at https://rpcec.sld.cu. The study adhered to Good Clinical Practice (GCP) guidelines and received approval from multiple institutional and governmental ethics boards, as detailed in the manuscript.

-Is the "Informed Consent Statement" form available as supplementary material?

RESPONSE: We appreciate the reviewer’s attention to ethical transparency. As requested by the Editor of Microorganisms, we provided the informed consent form in both Spanish and English translation. Should it be deemed necessary for full transparency, we would be glad to include the informed consent form as supplementary material in the final publication.

-Did the authors also use a microbial-culture dependent approach? It would be interesting to know which microorganisms are viable at each time point.

RESPONSE: We thank the reviewer for this insightful observation. In this study, we did not employ a microbial culture–dependent approach, as the primary objective was to assess shifts in gut microbial composition and predicted functionality using culture-independent methods, specifically 16S rRNA gene sequencing. This approach was selected to enable comprehensive profiling of the microbiome, including taxa that are difficult to culture or currently considered unculturable, as well as those in a viable but non-culturable (VBNC) state. While determining the viability of individual taxa across time points is indeed of scientific interest, it was beyond the scope of the current investigation. We agree that future studies incorporating both culture-based and molecular approaches could yield complementary insights into the viability and metabolic activity of key microbial populations modulated by SLE-F supplementation.

Reviewer 2 Report

Comments and Suggestions for Authors

This manuscript investigates the gut microbiome modulation and health benefits of a novel fucoidan extract from Saccharina latissima through a double-blind, placebo-controlled trial. Overall, the experimental design is reasonable, and the research methods are feasible. The results provide theoretical reference value for the clinical application of fucoidan extract in improving gut microbiome modulation and health benefits. However, there are several issues that require further clarification and revision by the authors:

Line 281: The primer sequences should be provided in the 5’ to 3’ direction.

Abbreviations in tables: All abbreviations appearing in the results tables (e.g., RBC, HTC in Table 4) should be fully explained in the table notes.

Lines 122–144: The preclinical studies, including in vitro microbial fermentation assays and Caco-2 cell experiments, are described in the Materials and Methods section but are not presented in the Results section. These results should be supplemented rather than simply stating “data not shown.”

Table 4: For the blood biochemical parameters, baseline values at the start of the trial should be provided for comparison.

Discussion section: The current discussion is overly extensive and should focus more on the key findings. Additionally, a comprehensive discussion integrating multiple indicators is recommended.

Methodological details: The manuscript lacks detailed descriptions of several assay methods, including SCFA and gas production measurements, in vitro microbial fermentation assays, and the Caco-2 cell experiment protocol.

Blood sample processing: It should be clearly stated whether serum or plasma samples were used for the final analysis.

Lines 261–264: Since EDTA anticoagulant tubes were used for blood sample collection, the supernatant obtained after centrifugation should be plasma, not serum. This needs to be corrected.

Microbiome analysis: Please supplement the PCoA (Principal Coordinate Analysis) results for the microbiome data.

Abbreviations: The first occurrence of each abbreviation (e.g., BCFA) should be defined. Please check the entire manuscript and make the necessary revisions.

Participant diet: Was the dietary intake of participants monitored during the trial? Dietary habits could significantly influence the results and should be addressed.

Author Response

REVIEWER 2

This manuscript investigates the gut microbiome modulation and health benefits of a novel fucoidan extract from Saccharina latissima through a double-blind, placebo-controlled trial. Overall, the experimental design is reasonable, and the research methods are feasible. The results provide theoretical reference value for the clinical application of fucoidan extract in improving gut microbiome modulation and health benefits. However, there are several issues that require further clarification and revision by the authors:

Line 281: The primer sequences should be provided in the 5’ to 3’ direction.

RESPONSE: We thank the reviewer for this helpful suggestion. The primer sequences have now been updated to be explicitly presented in the 5’ to 3’ direction, as recommended. This correction is reflected in the revised version of the manuscript at line 281.

The primer sequences are now represented as: Illumina F: 5´CCTACGGGNGGCWGCAG3´ and IlluminaR: 5´GACTACHVGGGTATCTAATCC3´.

Abbreviations in tables: All abbreviations appearing in the results tables (e.g., RBC, HTC in Table 4) should be fully explained in the table notes.

RESPONSE: We thank the reviewer for this valuable observation. The abbreviations appearing in the results tables, including RBC, HTC, and others, have now been fully defined in the corresponding table footnotes in the revised manuscript, as suggested. This change ensures greater clarity and consistency for readers.

RBC: Blood Cells (red blood cell count) HTC: Hematocrit MCV: Mean Corpuscular Volume MCH: Mean Corpuscular Hemoglobin MCHC: Mean Corpuscular Hemoglobin Concentration RDW-SD: Red Cell Distribution Width – Standard Deviation RDW-CV: Red Cell Distribution Width – Coefficient of Variation MPV: Mean Platelet Volume CRP: C-Reactive Protein ALAT: Alanine Aminotransferase ASAT: Aspartate Aminotransferase WBC: White Blood Cells HbA1c: Glycated Hemoglobin

Lines 122–144: The preclinical studies, including in vitro microbial fermentation assays and Caco-2 cell experiments, are described in the Materials and Methods section but are not presented in the Results section. These results should be supplemented rather than simply stating “data not shown.”

RESPONSE:  We thank the reviewer for this important observation. In response, we have substantially expanded the Results section to include a summary of the key findings from the in vitro fermentation and Caco-2/THP1 assays. Specifically, we now report results on short-chain fatty acid production, gas pressure, microbial taxonomic shifts, and metabolite profiling relevant to cardiovascular health. Additionally, we describe outcomes from the epithelial integrity and inflammation assays using the Caco-2/THP1 co-culture model.

To maintain clarity and conciseness in the main text, full methodological details, statistical outputs, and graphical data have been transferred to Supplementary Material S1, as now referenced in both the Methods and Results sections. We have removed all instances of “data not shown” to ensure transparency.

Table 4: For the blood biochemical parameters, baseline values at the start of the trial should be provided for comparison.

RESPONSE: We thank the reviewer for this important suggestion. Baseline (pre-intervention) values for the blood biochemical parameters were collected and used to calculate the within-subject change scores (Δ0–90), which are presented in Table 4 to highlight intervention-associated effects while controlling for inter-individual variability.

To improve interpretability, we have now modified Table 4 to include the baseline mean ± SD for each parameter across all groups. Additionally, we have included a supplementary table (Supplementary Table S2) providing the full baseline dataset by group for reference, as recommended. These additions aim to enhance clarity and facilitate comparisons across timepoints and study arms.

Discussion section: The current discussion is overly extensive and should focus more on the key findings. Additionally, a comprehensive discussion integrating multiple indicators is recommended.

RESPONSE:

Thank you for your incisive comments We appreciate the reviewer’s constructive suggestion. In response, we have substantially revised the Discussion to reduce redundancy, streamline content, and emphasize key findings. Specifically:

- Sections 4.3.2 and 4.3.7 were consolidated into a single, focused subsection (“Microbial Health and Functional Implications”) to avoid repetition.

-We integrated microbiome, functional, and host-level indicators (e.g., SCFA-producing taxa, inflammatory taxa, glucose and uric acid trends) to present a more comprehensive and mechanistically coherent interpretation of the results.

-We added bridging sentences to explicitly connect microbiome shifts to observed clinical or metabolic parameters (e.g., uric acid, glucose).

-A new synthesis paragraph was added at the end of Section 4.4 to summarize and contextualize the differential effects of high-dose and low-dose SLE-F, reinforcing broader clinical and research relevance.

Methodological details: The manuscript lacks detailed descriptions of several assay methods, including SCFA and gas production measurements, in vitro microbial fermentation assays, and the Caco-2 cell experiment protocol.

RESPONSE: We thank the reviewer for this helpful suggestion. To address this point, we have revised the relevant sentence in the Methods section to explicitly state the type of biological sample used. The revised text now reads:

“Blood samples were collected via venipuncture of the cubital vein. Serum was isolated following centrifugation and used for all biochemical and inflammatory marker analyses, including C-reactive protein (CRP), measured using the Cobas 6000 immunochemical analyzer (Roche Diagnostics, Rotkreuz, Switzerland).”

Blood sample processing: It should be clearly stated whether serum or plasma samples were used for the final analysis.

Lines 261–264: Since EDTA anticoagulant tubes were used for blood sample collection, the supernatant obtained after centrifugation should be plasma, not serum. This needs to be corrected.

RESPONSE: We thank the reviewer for this important clarification. The EDTA anticoagulant tubes referenced in this section were used specifically for hematological analyses and HbA1c measurement using the Cobas analyzer, which processes whole blood directly. These samples were not centrifuged for plasma or serum isolation. We have updated the manuscript to clarify that the EDTA tubes supported whole-blood analysis rather than plasma collection, thereby ensuring consistency with the laboratory protocols used during the study.

Microbiome analysis: Please supplement the PCoA (Principal Coordinate Analysis) results for the microbiome data.

RESPONSE: We thank the reviewer for this suggestion. While Principal Coordinate Analysis (PCoA) was not included in this study, we employed a supervised multivariate discriminant analysis to assess patterns of separation among treatment groups based on clinical and biochemical markers. As described in Section 3.4 and shown in Figure 2, this approach successfully classified 100% of subjects into their respective groups using two discriminant functions. Although this method was applied to clinical rather than microbiome data, it provided complementary insights into treatment-specific biological responses. We agree that PCoA or other beta-diversity visualizations of microbiome data would enhance future work and plan to include such analyses in subsequent studies.

Abbreviations: The first occurrence of each abbreviation (e.g., BCFA) should be defined. Please check the entire manuscript and make the necessary revisions.

RESPONSE: We thank the reviewer for this important observation. We have carefully reviewed the manuscript and ensured that all abbreviations, including BCFA (branched-chain fatty acids), are defined at their first mention. This revision has been applied consistently throughout the text for clarity and reader accessibility.

Participant diet: Was the dietary intake of participants monitored during the trial? Dietary habits could significantly influence the results and should be addressed.

RESPONSE: We thank the reviewer for this important observation.  While individual dietary intake was not actively monitored throughout the trial, all participants shared a common dietary background based on the traditional Cuban diet, which is characterized by high intake of plant-based staples, moderate consumption of animal protein, and limited processed food availability. This dietary pattern has been previously described in the literature and was considered a stable background variable across the study population. This context is also discussed in the published article (An Acad Cienc Cuba. 2025;15(1)) as part of the rationale for conducting microbiome research in this population. We acknowledge the influence of diet on gut microbiota and have included a clarifying statement in the revised manuscript to reflect this consideration.

Reviewer 3 Report

Comments and Suggestions for Authors

The per-reviewed manuscript is addressed to the current issues related to the study of the effects of a bioactive fucoidan-rich extract on gut microbial composition and functionality, as well as on the overall health of humans. The manuscript is properly structured, including all the necessary sections; each section is clearly written and easily understandable. The text is written in a clear and concise manner. The manuscript provides valuable insight for experts in microbiology, biotechnology, and genetics. Overall, the manuscript has received positive feedback. However, there are some questions for the author, as noted below

At the end of the Introduction section, clearly state the main purpose and objectives of your research.

The manuscript describes in great detail the change in the taxonomic composition of the gut microbiome, and the authors also provide statistical analysis of the data obtained. However, I suggest to replace some of the significant amount of values ​​presented at the article to the supplementary materials

Author Response

REVIEWER 3

The per-reviewed manuscript is addressed to the current issues related to the study of the effects of a bioactive fucoidan-rich extract on gut microbial composition and functionality, as well as on the overall health of humans. The manuscript is properly structured, including all the necessary sections; each section is clearly written and easily understandable. The text is written in a clear and concise manner. The manuscript provides valuable insight for experts in microbiology, biotechnology, and genetics. Overall, the manuscript has received positive feedback. However, there are some questions for the author, as noted below

At the end of the Introduction section, clearly state the main purpose and objectives of your research.’’

RESPONSE: We thank the reviewer for this helpful suggestion. In response, we have ensured that the final paragraph of the Introduction explicitly states the main purpose and objectives of the study. This section now clearly highlights the focus on assessing the impact of SLE-F on microbial diversity, taxonomic shifts, and predicted functional outcomes, as well as the broader clinical implications.

The manuscript describes in great detail the change in the taxonomic composition of the gut microbiome, and the authors also provide statistical analysis of the data obtained. However, I suggest to replace some of the significant amount of values ​​presented at the article to the supplementary materials

RESPONSE: We appreciate the reviewer’s thoughtful suggestion. However, we have elected to retain the key data within the main body of the manuscript to preserve the narrative flow and facilitate interpretation by the reader without the need to consult supplementary materials. Additionally, other reviewers requested more integrated data presentation (e.g., PCoA or composite analyses), reinforcing the value of having essential results accessible in the main text. We have made efforts to streamline the figures and tables for clarity and brevity, while maintaining scientific transparency and coherence.

Round 2

Reviewer 2 Report

Comments and Suggestions for Authors

The author has made revisions to my feedback.